# DETR-ViP: Detection Transformer with Robust Discriminative Visual Prompts

**Bo Qian**[1,2*]  **Dahu Shi**[3,4]  **Xing Wei**[1,2†]

[1] School of Software Engineering, Xi'an Jiaotong University
[2] State Key Laboratory of Human-Machine Hybrid Augmented Intelligence, Xi'an Jiaotong University
[3] CCAI, Zhejiang University    [4] Hikrobot Co., Ltd.

## Abstract

Visual prompted object detection enables interactive and flexible definition of target categories, thereby facilitating open-vocabulary detection. Since visual prompts are derived directly from image features, they often outperform text prompts in recognizing rare categories. Nevertheless, research on visual prompted detection has been largely overlooked, and it is typically treated as a byproduct of training text prompted detectors, which hinders its development. To fully unlock the potential of visual-prompted detection, we investigate the reasons why its performance is suboptimal and reveal that the underlying issue lies in the absence of global discriminability in visual prompts. Motivated by these observations, we propose DETR-ViP, a robust object detection framework that yields class-distinguishable visual prompts. On top of basic image-text contrastive learning, DETR-ViP incorporates global prompt integration and visual-textual prompt relation distillation to learn more discriminative prompt representations. In addition, DETR-ViP employs a selective fusion strategy that ensures stable and robust detection. Extensive experiments on COCO, LVIS, ODinW, and Roboflow100 demonstrate that DETR-ViP achieves substantially higher performance in visual prompt detection compared to other state-of-the-art counterparts. A series of ablation studies and analyses further validate the effectiveness of the proposed improvements and shed light on the underlying reasons for the enhanced detection capability of visual prompts.

## 1 Introduction

Compared with traditional closed-set object detection( Girshick (2015); Redmon et al. (2016); Zhang et al. (2022a)), open-set object detection breaks through the limitation of predefined categories, thereby demonstrating greater potential in real-world applications. Prompt-based object detection has further inspired the development of open-set detection by enabling flexible identification of target objects. Several prominent approaches( Ghiasi et al. (2022); Gu et al. (2021); Kamath et al. (2021); Li et al. (2022b); Liu et al. (2024); Minderer et al. (2022); Zhou et al. (2022)) support text-prompted open-set detection by distilling knowledge from vision-language models such as CLIP( Radford et al. (2021)) or from language models like BERT( Devlin et al. (2019)) through aligning visual representations with textual descriptions. The training paradigm of text-prompted model centers on aligning visual and textual feature spaces, with zero-shot generalization largely attributed to large-scale, generic image-text pairs. However, in specialized domains, relying solely on textual descriptions of target categories or attributes often proves insufficient for reliable detection.

Some methods( Jiang et al. (2023); Kirillov et al. (2023); Zou et al. (2023); Ravi et al. (2024); Li et al. (2024)) employ visual prompts to define target objects of interest, further enhancing the interactivity and adaptability of object detection models. Recently, several methods( Jiang et al. (2024); Cheng et al. (2024)) have begun to explore object detection frameworks that support both textual and visual prompts, achieving promising progress. Jiang et al. (2024) empirically find that, compared with text prompts, visual prompts are more effective in recognizing rare categories. This is

---

*Work done during the internship at Hikrobot Co., Ltd.    †Corresponding author.
‡Project page: https://github.com/MIV-XJTU/DETR-ViP

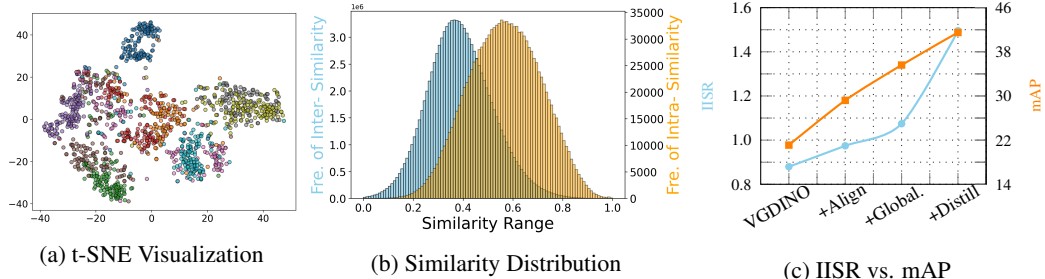

(a) t-SNE Visualization

(b) Similarity Distribution

(c) IISR vs. mAP

Figure 1: **Analysis of visual prompts.** (a) t-SNE visualization of VIS-GDINO prompts sampled from 10 COCO categories. (b) Similarity distribution between VIS-GDINO prompts of the same category and across different categories. (c) Trends of Intra-Inter Similarity Ratio (IISR) and mAP. expected because visual prompts, being sampled from the visual domain, are naturally compatible with image features, thus possessing **stronger generalization ability** than text prompts, which require cross-modal (text-to-vision) alignment. Nevertheless, visual prompts still underperform text prompts overall, which limits their practical applicability.

To systematically investigate the reasons behind this suboptimal performance and enhance detection capability, we build a baseline model incorporating visual prompts on top of Grounding DINO, termed VIS-GDINO, and conduct a detailed analysis. VIS-GDINO achieves mAP scores of 21.1 and 17.2 on COCO and LVIS, respectively, which are lower than those of Grounding DINO with text prompts. For each category, we sample 128 images to extract visual prompts, and perform a t-SNE analysis on all visual prompts, as shown in Figure 1(a). It can be observed that the visual prompts of VIS-GDINO lack clear semantic boundaries, which may contribute to frequent misclassifications when using visual prompts for categorization. To illustrate this more clearly, we compute the pairwise cosine similarities among all visual prompts and plot the distributions of similarities for same-category and different-category prompts as histograms, shown in Figure 1(b). Same-category and different-category prompts share a relatively wide similarity range, which underscores the disorganization of the prompt embedding space. Taken together with these observations, we attribute the suboptimal performance of visual prompts to two factors: (1) visual prompts from different instances of the same category exhibit large variance; and (2) visual prompts from different categories are heavily entangled in the global embedding space, making them difficult to distinguish. **In summary, visual prompts suffer from a lack of sufficient semantic discriminability.**

Building upon the above insight, we improve VIS-GDINO into a detection model with more robust and discriminative visual prompts, termed **DETR-ViP**, by simultaneously reducing the intra-class variance and enlarging the inter-class distance of visual prompts. DETR-ViP introduces a *global prompt integration* strategy that simulates cross-image prompt detection during training while augmenting negative samples, thereby encouraging image features to encode more distinctive global semantics. In addition, we design a *visual-text prompt relation distillation* loss. This loss transfers the semantic priors from the similarity matrix of textual prompts to the visual prompts, encouraging them to capture semantic proximity. Finally, we propose a *selective fusion* strategy. The fusion of each visual prompt is conditioned on the model's judgment of whether the corresponding category instance exists in the target image, resulting in more stable and robust detection. To quantitatively evaluate the quality of visual prompts by measuring intra-class variance and inter-class separability, we further introduce the Intra-Inter Similarity Ratio (IISR). A larger IISR indicates stronger semantic consistency of visual prompts. As shown in Figure 1(c), both IISR and mAP on COCO consistently improve with the proposed modifications, validating that the performance gains stem from the optimization of visual prompt distributions.

We conduct comprehensive evaluations on COCO( Lin et al. (2014)), LVIS( Gupta et al. (2019)), ODinW( Li et al. (2022a)), and Roboflow 100( Ciaglia et al. (2022)). Under comparable backbone settings, DETR-ViP consistently outperforms T-Rex2. Specifically, DETR-ViP-T outperforms T-Rex2-T by +4.4 mAP on COCO, and achieves +3.7 mAP over T-Rex2-T and +6.9 mAP over YOLOE-v8-L on LVIS. In terms of $AP_c$ and $AP_r$, our model surpasses T-Rex2 by +9.4 and +5.2, and exceeds YOLOE-v8-L by +8.7 and +1.9, respectively. On ODinW and Roboflow 100, which contain category distributions substantially different from the training datasets, DETR-ViP-T surpasses T-Rex2-L by 3.4 and 5.1 mAP, respectively. DETR-ViP-L further improves upon DETR-ViP-T, surpassing T-Rex2-L by 3.7 and 1.5 mAP on COCO and LVIS, respectively. Beyond fair comparisons

with existing methods, we also conducted extensive ablation studies to validate the effectiveness of each modification from VIS-GDINO to DETR-ViP. Moreover, both the IISR metric and t-SNE visualizations of visual prompts provide intuitive evidence that these improvements progressively optimize the structure of the prompt embedding space.

## 2 RELATED WORK

Prompt-based object detection provides a promising pathway toward open-vocabulary detection. Text-prompted approaches( Gu et al. (2021); Kamath et al. (2021); Li et al. (2022b); Liu et al. (2024); Minderer et al. (2022); Zang et al. (2022); Zhang et al. (2023); Zhong et al. (2022)) have achieved remarkable progress, exhibiting strong zero-shot and few-shot recognition capabilities. These methods mainly align visual features with text embeddings, often leveraging pretrained language models such as CLIP( Radford et al. (2021)) or BERT( Devlin et al. (2019)). Representative advances include GLIP( Li et al. (2022b)), which unifies detection and phrase grounding through large-scale image–text pretraining; DetCLIP( Yao et al. (2022)), which introduces richly descriptive concepts; Grounding DINO( Liu et al. (2024)), which enhances early cross-modal fusion; and RegionCLIP( Zhong et al. (2022)), which transfers regional knowledge via pseudo boxes to improve generalization.

However, in many scenarios language alone is insufficient to precisely describe target objects, motivating research on visual-prompted detection for greater flexibility and contextual awareness. Early methods( Minderer et al. (2022); Xu et al. (2023); Zang et al. (2022); Jiang et al. (2023)) adopted full reference images as prompts, while later works( Kirillov et al. (2023); Ravi et al. (2024); Li et al. (2024)) explored more compact formats such as keypoints or bounding boxes. DINOv( Li et al. (2024)) treats visual prompts as contextual exemplars for open-vocabulary detection, and T-Rex( Jiang et al. (2023)) applies deformable attention to extract region-level features from point- or box-based prompts. Recent methods combine visual and textual prompts: T-Rex2( Jiang et al. (2024)) incorporates vision–language contrastive learning, and YOLOE( Cheng et al. (2024)) introduces RepRTA and SAVPE for unified prompt handling. Despite these advances, the performance of visual-prompted detection still lags behind that of text-prompted counterparts.

## 3 METHOD

### 3.1 PRELIMINARY

A closed-set detector computes the score of each proposal for predefined categories by training a linear layer, whereas an prompt-based open-set detector derives the score by measuring the similarity between the proposal features $O$ and the prompt embeddings $P$, as formulated in Equation (1).

$$\underset{\text{Closed-set}}{\text{Score} = \sigma(OW^\top + b)} \quad \longleftrightarrow \quad \underset{\text{Prompt-based}}{\text{Score} = \sigma(OP^\top + b),} \tag{1}$$

Here, $W$ denotes the weights of the linear layer, $b$ is a learnable bias, and $\sigma$ represents the sigmoid function. In this work, we adopt the prompt-based method shown in Equation (1) (right), where neither $O$ nor $P$ is subjected to $L_2$ normalization.

Given an image-text pair $(I, T)$, Grounding-DINO( Liu et al. (2024)) extracts multi-scale visual features $X_I \in \mathbb{R}^{L \times C}$ using a visual backbone (*e.g.*, Swin Transformer( Liu et al. (2021))), and obtains textual prompt embeddings $P_T \in \mathbb{R}^{N_t \times C}$ through a text backbone (*e.g.*, BERT( Devlin et al. (2019))). Subsequently, $X_I$ and $P_T$ are fused and enhanced within the encoder, whose layers consist of a deformable attention module (for enhancing $X_I$), a self-attention module (for enhancing $P_T$), and a bidirectional attention module (for fusing $X_I$ and $P_T$), as shown in Equation (2). Similar modules are also present in the decoder. Given $K$ user-specified normalized boxes $b_j = (x_j, y_j, w_j, h_j), j \in \{1, 2, ..., K\}$, T-Rex( Jiang et al. (2023)) first extracts multi-scale features $X_I$ using a visual backbone and encoder. Each box is transformed into a positional embedding via a sine-cosine encoding and projected to a uniform dimension to obtain box embeddings $B$. A learnable content embedding $C$ is replicated $K$ times and concatenated with $B$, together with a global box $B' = \{0.5, 0.5, 1, 1\}$ and a class token $C'$ for aggregating visual prompt features. The resulting embeddings are linearly mapped to form the visual prompt queries $Q$, which are then fed into a multi-scale deformable cross-attention module to extract visual prompts $P_V$ from $X_I$. The overall process is summarized in Equation (3).

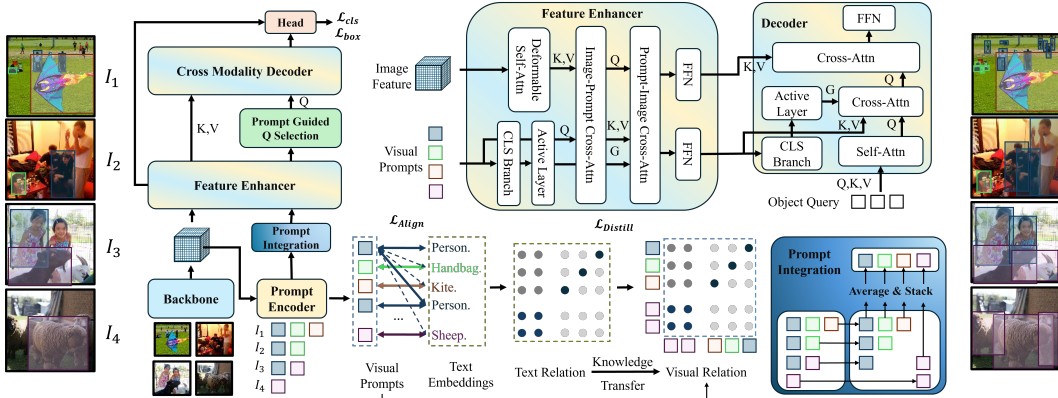

Figure 2: **The overview of DETR-ViP.** DETR-ViP builds on Grounding DINO by incorporating a visual prompt encoder for visual-prompted detection. It improves prompt semantics via global prompt Integration and visual-textual prompt relation distillation, and refines the fusion module to stabilize image-prompt interactions, thereby enhancing detection robustness.

$$\begin{cases} X_I = \text{MSDeformSelfAttn}(X_I) \\ P_T = \text{SelfAttn}(P_T) \\ X_I, P_T = \text{BiAttn}(X_I, P_T) \end{cases}, \quad (2) \qquad \begin{cases} B = \text{Linear}(\text{PE}(b_1, b_2, ..., b_K)) \\ Q = \text{Linear}(\text{CAT}([C; C'], [B; B'])) \\ P_V = \text{MSDeformAttn}(Q, B, X_I) \end{cases}. \quad (3)$$

They both adopt the same detection loss as DINO Zhang et al. (2022a), which consists of a classification loss $\mathcal{L}_{\text{cls}}$ and a regression loss composed of an L1 loss $\mathcal{L}_1$ and a GIoU loss $\mathcal{L}_{\text{GIoU}}$, as well as a denoising loss $\mathcal{L}_{\text{dn}}$. T-Rex2 further employs an image-text contrastive loss $\mathcal{L}_{\text{Align}}$. We also adopt these losses, which we denote as $\mathcal{L}_{\text{base}}$:

$$\mathcal{L}_{\text{base}} = \lambda_{\text{cls}}\mathcal{L}_{\text{cls}} + \lambda_1\mathcal{L}_1 + \lambda_{\text{GIoU}}\mathcal{L}_{\text{GIoU}} + \lambda_{\text{dn}}\mathcal{L}_{\text{dn}} + \lambda_{\text{Align}}\mathcal{L}_{\text{Align}}, \quad (4)$$

Here, $\lambda_\bullet$ denotes the weighting coefficients for each loss term.

### 3.2 DETR-VIP

We develop the baseline VIS-GDINO from Grounding DINO by inserting the visual prompt encoder, as defined in Equation (3), between the backbone and the encoder, and removing the fusion modules in the encoder and decoder as represented in Equation (2). On top of this architecture, we introduce the *global prompt integration*, *visual-textual prompt relation distillation loss*, and *selective fusion* strategy to enhance visual prompt detection, thereby upgrading VIS-GDINO to DETR-ViP, as shown in Figure 2.

**Global Prompt Integration.** DETR-like models typically optimize confidence scores using Focal Loss ( Lin et al. (2017)) to endow the model with classification capability. In prompt-based detection, given a proposal $q$, suppose it is matched to the category represented by the prompt $p^+$, while $p^-$ denotes the prompts from other categories. The classification loss contributed by this proposal is formulated as follows:

$$-\alpha\Big[\underbrace{(1 - q \cdot p^+)^\gamma \log q \cdot p^+}_{\text{positive item}} + \underbrace{\sum (q \cdot p^-)^\gamma \log(1 - q \cdot p^-)}_{\text{negative items}}\Big]. \quad (5)$$

The positive term attracts proposal features toward positive prompt embeddings, while the negative term repels them, akin to contrastive learning. Prior work ( Chen et al. (2020); He et al. (2020)) shows that abundant negatives are essential for learning globally optimal representations. However, training samples in each iteration typically contain only a small number of categories, and under the "current image prompt, current image detect" strategy in T-Rex2 — where prompts are sampled exclusively from the GT boxes of the current training image — the classification problem degenerates into a very small $N$-way classification task. As a result, the limited category diversity directly constrains the model's global discriminability.

For textual prompts, this can be easily achieved by padding them with additional category phrases to a fixed length. The extension of visual prompts is more challenging, as the extraction of visual

prompts relies on the images they originate from. Sampling an additional batch of images during training to extract negative examples would significantly reduce training efficiency. Therefore, we adopt a strategy that aggregates prompts from all samples and integrates them into a unified classifier. Specifically, we collect all visual prompts from the images within the same batch and group them by category. For each category, we compute the mean of all its prompts to obtain a class prototype. The prototypes of all categories are then concatenated and shared across all images in the current batch, serving as the weights of the classifier. Unlike the "current image prompt, current image detect" strategy, this strategy not only increases the number of negative examples for visual prompts, thereby stabilizing training, but also implicitly simulates cross-image prompting. This is because the visual prompts for a given category are aggregated not only from the current sample but also from positive examples in other samples. This simple strategy substantially improves the performance of detection based on visual prompts, as evidenced by the experiments reported in Section 4.4.

**Visual-Textual Prompt Relation Distillation.** Prompt-based detection localizes and classifies targets by selecting proposals with the strongest responses to the prompt, where the prompt functions analogously to a classifier. The prompt embedding space is expected to exhibit intra-class compactness and inter-class separability, with similar concepts clustering together and dissimilar ones remaining apart. Language encoders like CLIP and BERT inherently possess these properties thanks to their learning objectives: BERT's masked language modeling, which clusters semantically related words, and CLIP's contrastive pretraining, which aligns and separates concepts across modalities. However, since the appearance of instances can vary significantly due to individual differences, environmental factors, and other conditions, the distribution of visual prompts tends to exhibit high variance and blurred, ambiguous semantic boundaries. Multimodal detectors such as T-Rex2 address this issue by aligning visual prompts with their corresponding textual prompts. If such alignment were to be fully achieved, the visual prompt space would naturally inherit the semantic organization of the text embedding space, thereby obtaining strong intra-class compactness and inter-class discrimination. Nevertheless, studies ( Liang et al. (2022); Schrodi et al. (2024)) have shown that images and text cannot be perfectly aligned, which fundamentally limits the effectiveness of this indirect alignment approach. An intuitive approach is to apply the Supervised Contrastive Loss ( Khosla et al. (2020)) to visual prompt embeddings:

$$\mathcal{L}_{SCL} = \sum_{i \in S} \frac{-1}{|S^+(i)|} \sum_{j \in S^+(i)} \log \frac{\exp(p_i p_j / \tau)}{\sum_{k \in S} \exp(p_i p_k / \tau)}, \tag{6}$$

Here, $S^+(i)$ denotes the set of prompts that belong to the same category as $p_i$, while $S$ denotes the set of all prompts. However, such a hard contrastive loss treats all negative examples equally, making it incapable of capturing the correlations between concepts. Moreover, when training on grounding datasets, it may also suffer from the influence of false negatives. For instance, *women* and *person* are treated as different categories during training, yet considering their corresponding visual prompt embeddings as negatives against each other is clearly unreasonable. Considering that language models provide a strong prior for semantic similarity, we adopt the following visual-textual prompt relation distillation loss:

$$\mathcal{L}_{\text{distill}} = \text{CrossEntropy}(\text{Softmax}(CC^T), \text{Softmax}(PP^T)) \tag{7}$$

$$= \frac{-1}{N_q} \sum_{i=1}^{N_q} \sum_{j=1}^{N_q} \frac{exp(c_i c_j^T / \tau_t)}{\sum_k exp(c_i c_k^T / \tau_t)} \log \frac{exp(p_i p_j^T / \tau_v)}{\sum_k exp(p_i p_k^T / \tau_v)}, \tag{8}$$

Here, $N_q$ denotes the number of prompts, $p$ and $P$ represent the visual prompt embedding vector and matrix, respectively, $c$ and $C$ denote the corresponding textual feature vector and matrix, and $\tau_t$, $\tau_v$ are the temperature parameters. Note that both $c$ and $p$ represent L2-normalized vectors.

It is worth noting that, unlike the alignment loss which enforces constraints over visual-text pairs, the relation distillation loss leverages the relational structure among text-text pairs as a prior and directly optimizes the topology of the visual prompt space. This approach avoids the difficulties of cross-modal alignment and instead adjusts the interrelations among visual prompts in a more direct manner, enabling the learned visual prompt space to exhibit strong intra-class compactness and inter-class separability. Importantly, the two losses are compatible and complementary: the alignment loss guides visual prompts toward stable semantic anchors defined by text embeddings, while the relation distillation loss focuses on refining the structural relationships within the visual prompt space,

leading to a more discriminative and semantically consistent representation. Overall, the total loss of DETR-ViP is as follow:

$$\mathcal{L}_{\text{total}} = \mathcal{L}_{\text{base}} + \lambda_{\text{distill}}\mathcal{L}_{\text{distill}}. \tag{9}$$

**Selective Fusion.** Fusing text prompt embeddings with image features in the early stage is a commonly adopted technique in open-vocabulary object detection. Such fusion not only allows the prompt embeddings to capture image-specific information but also makes the regions in the image with high responses to the prompts more semantically salient, and is generally considered an effective means to enhance prompt-based detection performance. However, in practical applications, the provided prompts can be highly flexible. In interactive detection, users may only wish to detect objects of a few categories of interest within an image and thus provide only 1~2 prompts. In contrast, in batch annotation scenarios, users may specify a large number of prompts in advance (e.g., the 80 categories in COCO), while for a given image some of these categories may have no corresponding instances. Under such circumstances, it becomes necessary to consider whether fusing prompt embeddings of irrelevant categories could have negative effects.

To investigate this, we incorporate the fusion strategy of Grounding DINO into DETR-ViP, as formulated in Equation (2) The prompt-to-image fusion is defined in Equation (10), with image-to-prompt fusion obtained by swapping $X_I$ and $P_V$.

$$Q = W_Q X_I, K = W_K P_V, V = W_V P_V, X_I^o = \text{Softmax}(\frac{QK^T}{\sqrt{d}})V, \tag{10}$$

where $W_\square$ ($\square \in \{Q, K, V\}$) denote the query, key, and value projection matrices, respectively. $X_I^o$ represents the fused image features. We use the model equipped with this fusion strategy to perform detection with either a single prompt or all prompts, and visualize the corresponding results. The visualization results are shown in Figure 3. When all 80 COCO prompts are given, the model detects correctly, but performance collapses with only the '*person*' prompt, as the global prompt integration causes the model to overfit to many-prompt scenarios.

This phenomenon exposes the fragility of full fusion strategies. A robust fusion mechanism should yield stable detection regardless of the number of prompts. To this end, we propose a selective fusion strategy that first determines whether a category exists in the image, and fuses only the relevant prompts while suppressing the others. Concretely, we introduce a gating vector $G$ into the fusion layer's attention weights. The gate $g_c$ for a category $c$ is intended to satisfy: $g_c \to 0$ when $c$ is present in the image, and $g_c \to -\infty$ when $c$ is absent. To estimate category presence, an auxiliary classification branch computes the similarity matrix $S \in \mathbb{R}^{L_I \times N_P}$ between image features $X_I \in \mathbb{R}^{L_I \times D}$ and prompt embeddings $P_V \in \mathbb{R}^{N_P \times D}$. The maximum similarity $\text{MAX}(S, 0) \in \mathbb{R}^{N_P}$ for each prompt is taken as its confidence score, followed by a threshold activation function $\delta(\cdot)$, which outputs 0 if the input exceeds $\theta$ and $-\infty$ otherwise. The specific details of the selective fusion strategy are as follows:

$$X_I^o = \text{Softmax}(\frac{QK^T + G}{\sqrt{d}})V, G = \delta(\text{MAX}(S, \text{dim} = 0), \theta), S = \sigma(\text{Sim}(X_I, P_V) + b). \tag{11}$$

By applying the selective fusion strategy in both training and evaluation, the model filters out prompts with insufficient responses to the current image, leading to a more stable and robust fusion process.

## 4 EXPERIMENT

### 4.1 IMPLEMENTATION DETAILS

We provide two versions of our model based on Swin Transformer backbones: Tiny and Large. Both variants stack six Transformer encoder layers as the visual encoder, three deformable cross-attention layers as the visual prompt encoder, and six deformable cross-attention layers as the box decoder. We adopt AdamW( Loshchilov & Hutter (2017)) as the optimizer, with a learning rate of $1 \times 10^{-5}$ for the backbone and $1 \times 10^{-4}$ for the other modules. In Equations (4) and (9), $\lambda_{\text{cls}}, \lambda_1, \lambda_{\text{GIoU}}, \lambda_{\text{Align}}$, and $\lambda_{\text{distill}}$ are set to 1.0, 5.0, 2.0, 1.0, and 10.0, respectively. In Equation (8), $\tau_t$ and $\tau_v$ are set to 0.07 and 0.1. DETR-ViP is trained on both detection and grounding datasets, including Objects365 (V1)( Shao et al. (2019)) and GoldG( Kamath et al. (2021)) (comprising GQA( Hudson & Manning (2019)) and Flickr30k( Plummer et al. (2015))), with images from COCO( Lin et al. (2014)) excluded.

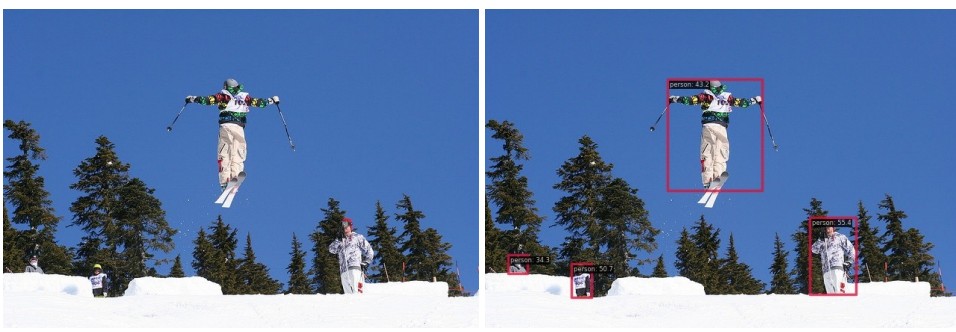

(a) Detect with '*person*' prompt          (b) Detect with all prompts

Figure 3: Illustration of Unstable Fusion. (a) With only the '*person*' prompt, detection fails. (b) With all 80 COCO category prompts, detection succeeds.

## 4.2 EVALUATION PROTOCOL AND METRICS

Following T-Rex2( Jiang et al. (2024)), we evaluate our method in the zero-shot setting on COCO( Lin et al. (2014)), LVIS( Gupta et al. (2019)), ODinW( Li et al. (2022a)), and Roboflow100( Ciaglia et al. (2022)), without training on any of these benchmarks. For fair comparison with T-Rex2( Jiang et al. (2024)) and YOLOE( Cheng et al. (2024)), evaluation is conducted under the Visual-G protocol. For fair comparison with T-Rex2 ( Jiang et al. (2024)) and YOLOE ( Cheng et al. (2024)), evaluation is conducted under two protocols: Visual-G and Visual-I. Under the Visual-G protocol, methods adopt a generic visual prompt workflow: for each benchmark, $N$ images per class are sampled from the training set, each containing at least one instance of the target class. All GT boxes of that class are used to extract the prompt embeddings, and their average serves as the visual prompt for evaluation. In contrast, under the Visual-I protocol, for a test image containing $M$ categories, only one GT box is randomly selected for each category to extract the corresponding visual prompt. We report standard AP on COCO; Fixed AP( Dave et al. (2021)), $AP_f$, $AP_c$, and $APr$ on LVIS, corresponding to frequent, common, and rare categories; and average AP across 35 ODinW and 100 Roboflow datasets.

To capture the semantic properties of visual prompts, we introduce **IISR** (I̲ntra-I̲nter S̲imilarity R̲atio) to quantify intra-class similarity and inter-class separability (Equation (12)). The numerator averages pairwise similarities within each category, while the denominator measures similarities among category-level mean vectors.

$$\text{IISR} = \frac{1}{|C|} \sum_{c \in C} \frac{1}{N_c(N_c - 1)} \sum_{i,j=1;i>j}^{N_c} \text{Sim}(p_i, p_j) \Bigg/ \frac{1}{|C|(|C| - 1)} \sum_{c,t=1;c>t}^{|C|} \text{Sim}(\tilde{p}_c, \tilde{p}_t) \quad (12)$$

Here, $C$ denotes the set of categories, $\text{Sim}(\cdot, \cdot)$ refers to the cosine similarity, $N_c$ indicates the number of prompts in category $c$, and $\tilde{p}_c, \tilde{p}_t$ represent the mean vectors of all prompts in categories $c, t$.

## 4.3 ZERO-SHOT OBJECT DETECTION

**Visual-G.** We evaluate the zero-shot generic detection capabilities of DETR-ViP on COCO, LVIS, ODinW, and Roboflow100, and report the results in Table 1. The results of T-Rex2 series and YOLOE series are quoted from their paper( Jiang et al. (2024); Cheng et al. (2024)). Experimental results show that DETR-ViP substantially outperforms YOLOE under the same training data. On LVIS, DETR-ViP-T surpasses YOLOE-v8-L by 6.9 AP, with gains of 1.9, 8.7, and 6.3 on $AP_r$, $AP_c$, and $AP_f$, respectively. With the same backbone, DETR-ViP also matches or exceeds T-Rex2 despite using far less data. For example, with Swin-T, DETR-ViP improves over T-Rex2 by 4.4 AP on COCO and by 3.7 AP on LVIS, with larger gains on $AP_r$ (+5.2) and $APc$ (+9.4). These improvements are most pronounced on common and rare categories, likely due to more efficient optimization of visual prompts, while advantages on frequent classes are smaller. Results on ODinW and Roboflow100, which exhibit larger domain shifts, further confirm this trend: DETR-ViP-T outperforms T-Rex2-L by 3.4 $APavg$ on ODinW and 5.1 on Roboflow100. DETR-ViP-L further improves over DETR-ViP-T, surpassing T-Rex2-L by 3.7 AP on COCO and 1.5 AP on LVIS. These results highlight the advantages of DETR-ViP compared with other visual prompt detection models.

Table 1: **Zero-shot generic detection evaluation on COCO, LVIS, ODinW, and Roboflow100.** For training data, O365, OI, HT, and CH indicate Object365( Shao et al. (2019)), OpenImages( Krasin et al. (2017)), HierText( Long et al. (2022)), and CrowdHuman( Shao et al. (2018)), respectively. GoldG( Kamath et al. (2021)) includes GQA( Hudson & Manning (2019)) and Flickr30k( Plummer et al. (2015)). Best results with Swin-T are underlined, and those with Swin-L are **boldfaced**.

| Model | Training Data | COCO AP | LVIS | | | | ODinW $AP_{avg}$ | Roboflow100 $AP_{avg}$ |
| | | | AP | $AP_r$ | $AP_c$ | $AP_f$ | | |
|---|---|---|---|---|---|---|---|---|
| T-Rex2-T | O365 OI,HT | 38.8 | 37.4 | 29.9 | 33.9 | 41.8 | 23.6 | 17.4 |
| T-Rex2-L | HT, CH SA-1B | 46.5 | 47.6 | 45.4 | 46.0 | **49.5** | 27.8 | 18.5 |
| YOLOE-v8-S | | - | 26.2 | 21.3 | 27.7 | 25.7 | - | - |
| YOLOE-v8-M | | - | 31.0 | 27.0 | 31.7 | 31.1 | - | - |
| YOLOE-v8-L | O365 | - | 34.2 | 33.2 | 34.6 | 34.1 | - | - |
| YOLOE-v11-S | GoldG | - | 26.3 | 22.5 | 27.1 | 26.4 | - | - |
| YOLOE-v11-M | | - | 31.4 | 27.1 | 31.9 | 31.7 | - | - |
| YOLOE-v11-L | | - | 33.7 | 29.1 | 34.6 | 33.8 | - | - |
| DETR-ViP-T | O365 | 43.2 | 41.1 | 35.1 | 43.3 | 40.4 | 31.2 | 23.6 |
| DETR-ViP-L | GoldG | **50.2** | **49.1** | **46.3** | **50.5** | 48.4 | **34.5** | **24.7** |

Table 2: **Zero-shot interactive detection evaluation on COCO and LVIS.**

| Model | COCO AP | LVIS | | | ODinW $AP_{avg}$ | Roboflow100 $AP_{avg}$ |
| | | AP | $AP_f$ | $AP_c$ | $AP_r$ | |
|---|---|---|---|---|---|---|---|
| T-Rex2-Swin-T | 56.6 | 59.3 | 54.6 | 63.5 | 64.4 | 37.7 | 30.6 |
| T-Rex2-Swin-L | 58.5 | 62.5 | 57.9 | 66.1 | 70.1 | 39.7 | 30.2 |
| DETR-ViP-T | 65.4 | 66.1 | 57.5 | 73.5 | 78.4 | 46.8 | 40.1 |
| DETR-ViP-L | 71.1 | 71.9 | 64.2 | 78.2 | 83.6 | 51.2 | 44.3 |

**Visual-I.** We evaluate the zero-shot interactive detection capability of DETR-ViP, and the results are presented in Table 2. Since YOLOE (Cheng et al. (2024)) does not report results under this setting, we compare only with T-Rex2 (Jiang et al. (2024)). Under the Visual-I protocol, categories that do not appear in the current image are excluded, which effectively provides prior knowledge about the image content and makes the task substantially easier than Visual-G detection. As shown in the results, the AP on COCO, LVIS, ODinW, and Roboflow100 are consistently higher under Visual-I than Visual-G. Moreover, DETR-ViP also surpasses T-Rex2 across the board under the Visual-I protocol. We attribute this to the presence of more negative examples during training, enabling DETR-ViP to learn a large N-way classification task, thereby strengthening its classification ability.

## 4.4 ABLATION EXPERIMENTS

DETR-ViP extends Grounding DINO( Liu et al. (2024)) by first introducing VIS-GDINO, a detector supporting visual prompts. Unlike Grounding DINO, VIS-GDINO removes fusion modules (Equation (2)) and inserts a visual prompt encoder (Equation (3)) between the backbone and encoder, enabling conditional detection based on user-provided reference boxes. We then progressively extend VIS-GDINO into DETR-ViP and validate each modification through ablation studies (Table 3). Furthermore, we sample 128 images for each category from the COCO training set and perform t-SNE visualization along with similarity analysis on the generated prompts, as shown in Figure 4.

**VIS-GDINO & Image-Text Align.** For the baseline VIS-GDINO, performance is limited, achieving only 21.1 mAP on COCO and 17.2 mAP on LVIS, as sampling a few prompts from the current image for classification training fails to learn a discriminative global distribution of visual prompt embeddings. Introducing the image-text contrastive loss substantially improves results to 29.2 mAP on COCO (+8.1) and 23.4 mAP on LVIS (+6.2), by distilling rich semantic priors from text features

Table 3: **Roadmap to DETR-ViP.** The left-aligned '+' indicates cumulative improvement, while the indented '+' marks a standalone improvement not accumulated into the subsequent variants.

| Model | COCO-val | | LVIS-minival | | | | |
|---|---|---|---|---|---|---|---|
| | AP | IISR | AP | $AP_f$ | $AP_c$ | $AP_r$ | IISR |
| VIS-GDINO-T | 21.1 | 0.8797 | 17.2 | 22.0 | 13.0 | 11.6 | 0.8954 |
| +Text-Image Alignment | 29.2 | 0.9743 | 23.4 | 27.8 | 19.5 | 18.4 | 0.9872 |
| +Global Prompt Integration | 35.6 | 1.0734 | 33.0 | 32.8 | 33.4 | 31.8 | 1.0550 |
| +Text-Region Distillation | 41.5 | 1.4954 | 39.5 | 39.0 | 41.5 | 33.4 | 1.2248 |
| +Encoder Fusion | 41.3 | 1.5001 | 39.1 | 38.4 | 41.3 | 33.1 | 1.2174 |
| +Encoder Selective Fusion | 42.2 | 1.4963 | 40.6 | 40.3 | 42.4 | 34.2 | 1.2165 |
| +Decoder Fusion | 40.8 | 1.4976 | 25.5 | 22.8 | 29.0 | 23.5 | 1.2172 |
| +Decoder Selective Fusion | 43.2 | 1.5010 | 41.1 | 40.4 | 43.3 | 35.1 | 1.2212 |

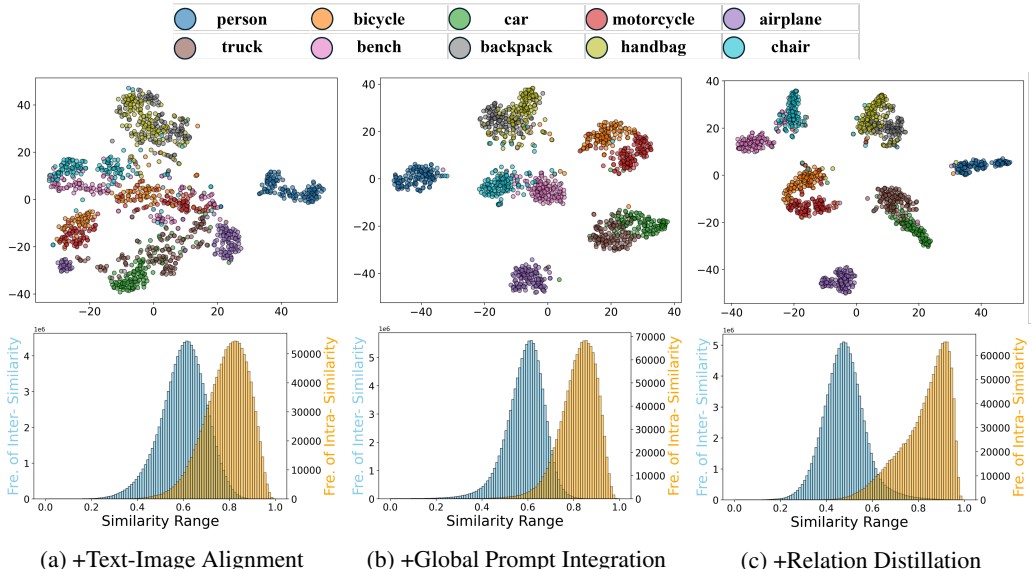

(a) +Text-Image Alignment  (b) +Global Prompt Integration  (c) +Relation Distillation

Figure 4: **Visual prompt analysis for different model variants.** (Top) t-SNE visualization of the visual prompts. (Bottom) Distribution of intra- and inter-class pairwise similarities.

into visual prompts. Analysis of visual prompts confirms this effect: in Figure 1(a), VIS-GDINO's prompts are scattered, whereas Figure 4(a, top) shows emerging cluster structures. Similarly, pairwise similarity distributions overlap heavily between intra- and inter-class prompts for VIS-GDINO (Figure 1(b)), but this overlap is greatly reduced after alignment (Figure 4(a, bottom)). In addtion, IISR increases by 0.0946 on COCO and 0.0918 on LVIS. These results indicate that the alignment enhances the semantic consistency of visual prompts, which in turn improves detection performance.

**Global Prompt Integration.** With global prompt integration, the intermediate DETR-ViP achieves 35.6 AP on COCO (+6.4) and 33.0 on LVIS (+9.6). This mechanism improves training efficiency by aggregating prompts of the same class across samples, which encourages intra-class clustering, while integrating prompts of different classes enlarges the negative set, sharpening inter-class boundaries. On LVIS, $AP_r$ and $AP_c$ increase markedly (+13.4 and +13.9), as rare and common categories can now contribute to the whole updates rather than only the current sample. As shown in Figure 4(b), visual prompts form clearer clusters with reduced intra-/inter-class overlap, confirming the effectiveness of global prompt integration in structuring the embedding space.

**Visual-Textual Prompt Relation Distillation.** As shown in Figure 4(c), introducing the visual-textual prompt relation distillation not only sharpens the inter-class boundaries but also reduces the intra-class variance of prompts. Although some categories exhibit closely clustered prompts, these

correspond to semantically related concepts such as truck and car, bench and chair, or backpack and handbag. This indicates that the prompt embedding space demonstrates semantic coherence, where semantically related concepts are mapped closer together and unrelated ones farther apart. The overlap between intra- and inter-class similarity distributions is further reduced, while the inter-class similarity distribution becomes more skewed toward 1.0, suggesting that intra-class prompts are nearly identical. On COCO and LVIS, IISR increased by 0.4220 and 0.1698, respectively. Benefiting from this, the intermediate variant of DETR-ViP achieved an additional AP gain of 5.9 on COCO, reaching 41.5, and 6.5 on LVIS, reaching 39.5. Specifically, $AP_r$, $AP_c$, and $AP_f$ improved by 1.6, 8.1, and 6.2, respectively.

**Selective Fusion.** The fusion module is designed to enable interactions between prompt embeddings and image features. Its primary role is to adapt the prompts to the characteristics of the current sample. Therefore, after introducing various fusion modules, the IISR of the unfused prompts remains almost unaffected. As shown in Table 3, directly adopting the encoder fusion from Grounding DINO brings almost no gains and even reduces AP on COCO and LVIS.

Practically, this naive strategy is highly sensitive to the number of prompts: detection works when all COCO categories are provided but fails with a single class prompt (Figure 3). As shown in Figure 5, experiments varying the number of prompts $N_p$ indicate that $AP_c$ increases with more prompts and gradually converges, revealing a train-test gap, since global prompt integration during training typically involves many prompts. The proposed selective fusion module first determines whether a category is present before interacting with its prompt, thereby suppressing irrelevant information and enabling robust detection. This approach stabilizes performance across different prompt numbers and improves AP by +0.7 on COCO and +1.1 on LVIS (Table 3). Applying full fusion to the decoder harms AP, as decoder fusion directly modifies object queries, amplifying the negative effect of irrelevant categories. When selective fusion is applied to the decoder as well, these issues are mitigated, yielding further gains of +1.0 AP on COCO and +0.5 on LVIS.

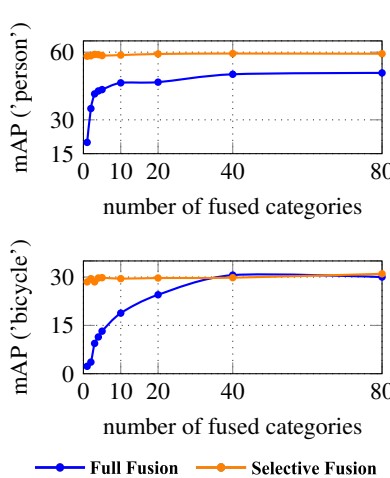

Figure 5: mAP *vs* num. of Prompts

## 5 CONCLUSION

We propose DETR-ViP, an open-vocabulary detection framework that substantially advances the baseline of visual-prompted detection. Building upon Grounding DINO, we construct a baseline model, VIS-GDINO, which supports visual prompts. Through this baseline, we identify that the suboptimal performance of visual prompts stems from the lack of semantic coherence. To this end, we introduce Global Prompt Integration and Visual-Textual Prompt Relation Distillation. Global Prompt Integration enhances the global semantic organization of the prompt space by increasing the number of positive and negative examples. Visual-Textual Prompt Relation Distillation transfers the relational priors encoded in textual prompts into visual prompts, thereby endowing them with structured semantic relationships. With these improvements, visual prompts not only reduce intra-class variance but also acquire sharper semantic boundaries, ultimately leading to enhanced detection performance. In addition, we refine the fusion module of Grounding DINO by introducing a Selective Fusion strategy, which enables DETR-ViP to perform stable and robust prompt fusion regardless of the number of user-provided prompts, thereby further improving detection performance. Extensive experiments demonstrating that under the Visual-G protocol, DETR-ViP significantly outperforms existing counterparts, validating its superior effectiveness. Moreover, comprehensive ablation studies confirm the contribution of each proposed component and provide in-depth analyses on how our method enhances the semantic organization of visual prompt distributions.

ACKNOWLEDGMENTS

This work was supported by the National Natural Science Foundation of China No. 62572385, the Fundamental Research Funds for the Central Universities No. xxj032023020, and CAAI-CANN Open Fund, developed on OpenI Community.

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

# A    STATEMENTS

**Use of Large Language Models (LLMs).**    During the preparation of this paper, we employed LLM-based tools to assist with writing and polishing. Specifically, we used the following instructions:

- "*Please act as a native English speaker and AI expert, and translate the following passage into English.*"

- "*Please help refine the following passage to make it more concise while preserving its meaning, and ensure it adheres to the style of AI conference papers.*"

The LLM was used solely for language assistance. All ideas, analyses, and experimental results are original contributions of the authors.

**Ethics Statement.**    This work does not involve human subjects, animals, or sensitive personal data. The study does not raise any ethical concerns regarding data collection, experimentation, or potential societal impacts.

**Reprodicibility Statement.**    To ensure reproducibility of our results, we commit to publicly releasing the code, pre-trained model weights, and training/evaluation logs necessary to reproduce the main findings reported in this paper. All resources will be provided with sufficient documentation to facilitate replication and verification by the research community.

# B    DETAILED MOTIVATION AND LIMITATION

**Motivation.**    The motivation behind this work is that visual prompts possess advantages not offered by text prompts, yet their potential remains underexplored. Compared with textual prompts, visual prompts are more effective for detecting rare categories. This conclusion has been validated in T-Rex2 ( Jiang et al. (2024)): as shown in Table 4, the AP of visual prompts on rare categories in LVIS, as well as on ODinW and Roboflow100, is generally higher than that of textual prompts.

Table 4: **Comparison between visual and text prompts of T-Rex2 on rare categories.** The results are taken from Table 1 of T-Rex2 ( Jiang et al. (2024)). In the LVIS column, the results on the left and right of "/" correspond to *minival* and *val*, respectively.

| Model | Prompt Type | LVIS($AP_r$) | ODinW($AP_{avg}$) | Roboflow100($AP_{avg}$) |
|---|---|---|---|---|
| T-Rex2-Swin-T | Text | $37.4/29.0$ | 18.0 | 8.2 |
| T-Rex2-Swin-T | Visual-G | $29.9_{(\downarrow 7.5)}/32.4_{(\uparrow 3.4)}$ | $23.6_{(\uparrow 5.6)}$ | $17.4_{(\uparrow 9.2)}$ |
| T-Rex2-Swin-L | Text | $49.2/42.7$ | 22.0 | 10.5 |
| T-Rex2-Swin-L | Visual-G | $45.4_{(\downarrow 3.8)}/43.8_{(\uparrow 1.1)}$ | $27.8_{(\uparrow 5.8)}$ | $18.5_{(\uparrow 8.0)}$ |

Jiang et al. (2024) also conducted a per-category accuracy comparison between visual and textual prompts on LVIS, as shown in Figure 4 of its original main paper. For frequent categories, text prompts outperform visual prompts in the majority of cases (Text:Visual = 254:151). However, for rare categories, the reverse is more common, with visual prompts achieving higher accuracy (Text:Visual = 84:253). This is because text prompts require aligning category "*concepts*" across the text and vision domains, which depends on large-scale vision-language pre-training; in contrast, visual prompts are directly sampled from the vision domain, and thus are inherently compatible and similar to the target regions. These observations highlight the necessity of studying visual prompts.

Although visual prompts perform well on rare categories, their performance on more common categories is significantly lower than that of text prompts, as shown in Table 5. We argue that this gap does not reflect the upper bound of visual prompts, but rather that their potential has not yet been fully explored. In this work, we aim to investigate the underlying reasons for the suboptimal performance of visual prompts and to fully exploit their potential, thereby advancing the development of prompt-based detection.

Table 5: **Comparison between visual and text prompts of T-Rex2 on frequent and common categories.** The results are taken from Table 1 of T-Rex2 Jiang et al. (2024). In the LVIS column, the results on the left and right of "/" correspond to *minival* and *val*, respectively.

| Model | Prompt Type | COCO AP | AP | LVIS AP$_f$ | AP$_c$ |
|---|---|---|---|---|---|
| T-Rex2-Swin-T | Text | 45.8 | $42.8/34.8$ | $46.5/41.2$ | $39.7/31.5$ |
| T-Rex2-Swin-T | Visual-G | $38.8_{(\downarrow 7.0)}$ | $37.4_{(\downarrow 5.4)}/34.9_{(\uparrow 0.1)}$ | $41.8_{(\downarrow 4.7)}/41.1_{(\downarrow 0.1)}$ | $33.9_{(\downarrow 5.8)}/30.3_{(\downarrow 1.2)}$ |
| T-Rex2-Swin-L | Text | 52.2 | $54.9/45.8$ | $56.1/50.2$ | $54.8/43.2$ |
| T-Rex2-Swin-L | Visual-G | $46.5_{(\downarrow 5.7)}$ | $47.6_{(\downarrow 7.3)}/45.3_{(\downarrow 0.5)}$ | $49.5_{(\downarrow 6.6)}/49.5_{(\downarrow 0.7)}$ | $46.0_{(\downarrow 8.8)}/42.0_{(\downarrow 1.2)}$ |

**Limitation.** While this work emphasizes the importance of visual prompts and primarily focuses on exploring their potential, it does not imply that we overlook the value of textual prompts. We fully recognize that textual prompts can more explicitly describe object attributes—such as color (e.g., "a woman in a red shirt") or spatial position (e.g., "a car on the left side of the image")—which are advantages that visual prompts inherently lack. Although this paper does not address textual prompts, we argue that combining more discriminative visual prompts with guided natural language prompts represents a promising direction toward more general open-world detection.

## C    MORE RELATED WORK: CONTRASTIVE LEARNING

Contrastive losses( Hadsell et al. (2006)) enable direct optimization of similarity within the representation space, eliminating the need to match inputs to fixed targets. This idea forms the foundation of many unsupervised representation learning methods( Wu et al. (2018); Oord et al. (2018); Hjelm et al. (2018); Chen et al. (2020); Zhuang et al. (2019); Tian et al. (2020); He et al. (2020)). InfoNCE( Oord et al. (2018)), one of the most widely used contrastive loss formulations, where decreasing the loss encourages the query $q$ to be similar to its positive key $k^+$ and dissimilar to negative keys $k^-$. Chen et al. (2020) introduces a self-supervised pretraining framework that treats two augmented views of the same image as a positive pair and different images as negative pairs, and learns representations using InfoNCE. He et al. (2020) emphasize the importance of large numbers of negatives and propose a memory bank mechanism to expand the negative pool. Although these approaches achieve remarkable progress in self-supervised visual pretraining, their inability to exploit the label information in annotated datasets limits their effectiveness on downstream tasks. To address this, Khosla et al. (2020) extend batch contrastive learning to the fully supervised setting, enabling the model to fully leverage category labels for contrastive representation learning.

Multimodal contrastive representation learning( Zhang et al. (2022b); Xu et al. (2021); Li et al. (2021); Jia et al. (2021)) aims to map inputs from different modalities into a shared embedding space, giving rise to a powerful form of weakly supervised pretraining. CLIP( Radford et al. (2021)), trained on 400 million image-text pairs from WebImageText, learns highly transferable visual representations and demonstrates strong zero-shot performance across diverse downstream tasks, making it a promising direction for open-set recognition. However, Liang et al. (2022) first identified the existence of a modality gap. Specifically, the image and text embeddings do not form a unified space but rather reside in two narrow cones separated by a gap, and the contrastive loss tends to preserve this geometric discrepancy. Schrodi et al. (2024) further observe that only a few embedding dimensions predominantly account for this modality difference, and there is no clear evidence that a larger gap correlates with improved downstream performance. Moreover, although simple post-hoc methods can shrink the gap geometrically, they generally fail to produce meaningful performance gains.

## D    DETAILS OF ARCHITECTURE

### D.1    BASELINE MODEL: VIS-GDINO

VIS-GDINO is a baseline model we built to support visual prompting. It can be regarded as inserting a visual prompt encoder between the backbone and the encoder of DINO, while replacing the classification head with the classifier shown in Equation (1)(right). Similarly, it can also be

interpreted as inserting a visual prompt encoder between the backbone and encoder of Grounding DINO, with the fusion modules in both the encoder and decoder removed.

As shown in Figure 6, given an input image $I$ and $K$ user-specified boxes $\{b_i\}_{i=1}^K$, VIS-GDINO first extracts image features $X$ through the backbone network. The visual prompt encoder then generates visual prompts $P_V$ from $X$ based on $\{b_i\}$, leveraging a deformable attention mechanism to effectively aggregate features within each box. The features $X$ are further encoded by a deformable encoder without any fusion modules to produce a set of proposals $O$. Scores are computed using the contrastive classification head shown on the right side of Equation (1) (i.e. Score $= \sigma(OP_V^\top + b)$), and the top-$k$ proposals are selected to initialize the decoder queries $Q$. Within the decoder, $Q$ passes through multiple layers composed of self-attention and deformable cross-attention modules, where the self-attention module models interactions among object queries and the cross-attention module extracts information from the image features. After decoding, $Q$ is multiplied with $P_V$ to compute the scores, and the predicted boxes are obtained via the box regression head.

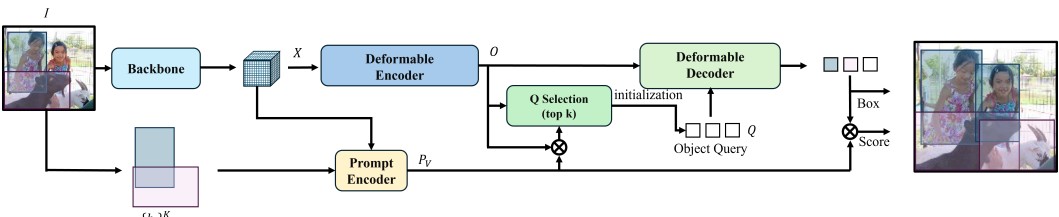

Figure 6: **A simplified illustration of VIS-GDINO.** Compared to Grounding DINO( Liu et al. (2024)), VIS-GDINO inserts a visual prompt encoder between the backbone and the encoder, and removes the fusion modules in both the encoder and the decoder.

### D.2 TEXT ENCODER

Unlike Grounding DINO, we use CLIP( Radford et al. (2021)) as the text encoder. We construct the input to the text encoder using the template "This is an image of ..." and take the pooled output as the text feature. To improve training efficiency, we pre-extract the text features with CLIP and store them for later use.

### D.3 FEATURE ENHANCER

The Feature Enhancer in DETR-ViP is built upon that of Grounding DINO, but with two key modifications: (i) the fusion module is replaced with the proposed selective fusion, and (ii) the self-attention layer over prompt embeddings is removed. This is because, unlike Grounding DINO which uses full sentences as text prompts and requires self-attention to extract features for each positive phrase, DETR-ViP represents each category with a single token. Specifically, for multiple box prompts of the same category within an image, the global token in the visual prompt encoder aggregates their information, while for global prompts across images, the average vector is used as the category-level prompt embedding.

### D.4 GLOBAL PROMPT INTEGRATION

This *Global Prompt Integration* operation is illustrated by the simple code snippet below.

```
prompts = gather(prompts) # N_i*d -> (N_1+N_2+...)*d
labels = gather(labels) # N_i -> (N_1+N_2+...)
gather_prompts = []
for l in labels.unique():
    gather_prompts.append(prompts[labels==l].sum(0) / (label==l).sum())
prompts = stack(gather_prompts, 0)
labels = labels.unique()
```

To intuitively illustrate how global prompt integration works, we provide an example: if the prompt categories in **sample1** are $C_1 = \{0, 2, 3, 5\}$ and those in **sample2** are $C_2 = \{0, 1, 4, 5\}$, then we

can use a classifier covering the combined category set $C = \{0, 1, 2, 3, 4, 5\}$ when performing classification across all samples. This approach not only increases the number of negative examples but also indirectly trains cross-image prompt detection, as the prompts for classes 0 and 5 from sample 1 are also used in the classification of sample 2.

For Visual Grounding (VG) datasets, the situation is typically more complicated. On the one hand, their annotations are generally of lower quality compared with object detection (OD) datasets such as COCO or Object365. On the other hand, the box descriptions in VG datasets are usually short phrases extracted from image captions, leading to highly variable formats. For example, a bounding box for a yellow scarf may be described as "blades to help board cut through water", which mixes action nouns with other nouns. Likewise, two boxes corresponding to two dogs may be described as "a short and white dog" and "two dogs". For such grounding data, integrating these phrases into prompts is challenging. If two descriptions are considered to belong to the same class only when their texts perfectly match, a large number of false negatives would be introduced, which in turn harms model training. To address this issue, we adopt two strategies. First, we decouple the training of OD and VG data. In each iteration, the entire batch is either from OD datasets or from VG datasets, thereby preventing noisy VG data from interfering with high-quality OD supervision. Second, for VG data, we extract the head noun using extract_head_noun(phrase) and obtain the lemma using get_lemma(phrase). Both operations are implemented with the spaCy library. In this manner, phrases such as "a short and white dog" or "two dogs" are normalized to the lemma "dog" before determining whether they belong to the same class. Although conceptually simple, this approach significantly reduces the occurrence of false negatives.

### D.5 CLASSIFICATION, ALIGNMENT, AND DISTILLATION LOSSES

For the classification loss, we follow T-Rex2 and Grounding DINO by computing confidence scores via matrix multiplication between proposal features and prompt embeddings, followed by a sigmoid activation, as defined in Equation (1). It's noted that both $O$ and $P$ are kept unnormalized, allowing their dot-product results to span the entire real domain $(-\infty, +\infty)$, which ensures that the sigmoid outputs are distributed within $(0, 1)$. For both the alignment and distillation losses, all vectors involved in the computation are L2-normalized

### E MORE EXPERIMENTS

**Downstream Transferring.** We conduct a downstream transfer experiment on COCO using the variant of DETR-ViP without the fusion module. Specifically, the class branch is replaced with a linear layer while all other modules are frozen, and the model is trained for 12 epochs. This setup is equivalent to fitting a classification hyperplane in the pre-learned feature space, thus serving as a measure of semantic consistency. As shown in Table 6, progressively adding image-text alignment, global prompt integration, and visual-textual distillation to VIS-GDINO steadily improves downstream AP, reflecting enhanced visual representations. However, the gains are smaller than those in Table 3, indicating that the main performance improvements arise from optimizing the visual prompt embedding space.

Table 6: **Downstream Transfer on COCO**

| Model | AP | $AP_{50}$ | $AP_{75}$ | $AP_S$ | $AP_M$ | $AP_L$ |
|---|---|---|---|---|---|---|
| VIS-GDINO-SwinT | 48.2 | 64.4 | 53.0 | 32.7 | 51.7 | 61.8 |
| +Text-Image Alignment | 48.5 | 65.3 | 53.3 | 32.3 | 52.0 | 62.6 |
| +Global Prompt Prompts | 48.8 | 65.8 | 53.5 | 33.8 | 52.4 | 62.7 |
| +Text-Region Distillation | 49.2 | 65.9 | 54.3 | 34.6 | 52.6 | 62.6 |

**Visual-Text Prompt Relateion Distillation *vs*. Supervised Contrastive Loss.** The supervised contrastive loss shown in Equation (6) is another option for enhancing the discriminability of visual features. In Table 7, we compare models trained with visual-textual prompt relation distillation versus those trained with supervised contrastive loss. It can be observed that the variant using supervised contrastive loss achieves AP scores that are 0.6 and 0.7 lower than DETR-ViP on COCO and LVIS,

respectively. We attribute this to the fact that supervised contrastive loss treats all negatives equally, which prevents visual prompts from establishing semantic coherence. Moreover, grounding datasets contain many synonyms and hierarchical relationships—for example, "human" and "person," or "human" and "gentle"—whose corresponding instances are visually from the same category. Pulling apart the embeddings of such semantically related prompts under supervised contrastive loss can therefore harm the optimization of the embedding space.

Table 7: **Comparison between Relation Distillation Loss and Supervised Contrastive Loss.**

| Model | COCO-val | LVIS-minival | | | |
|---|---|---|---|---|---|
| | AP | AP | $AP_f$ | $AP_c$ | $AP_r$ |
| $\mathcal{L}_{\text{Distill}}$ | 43.2 | 41.1 | 40.4 | 43.3 | 35.1 |
| $\mathcal{L}_{\text{SCL}}$ | 42.6 | 40.4 | 40.4 | 42.3 | 31.4 |

**Hyperparameter.** To ensure a fair comparison with other methods, we adopt the parameter settings of existing approaches for modules that are not our core contributions. Specifically, our $\lambda_{\text{cls}}, \lambda_1, \lambda_{\text{GIoU}}$, and $\lambda_{\text{dn}}$, are kept consistent with DINO, while $\lambda_{\text{Align}}$ follows the setting in T-Rex2. The focal loss hyperparameters $\gamma$ and $\alpha$ are set to 2.0 and 0.25, respectively, as commonly used in previous works. In this subsection, we perform ablation studies on the visual-textual relation distillation loss parameters $\lambda_{\text{distill}}, \tau_v$, and $\tau_t$, as well as the selective fusion threshold $\theta$.

First, we conduct an ablation study on $\lambda$. Table 8(a) reports the AP, $AP_r$, $AP_c$, and $AP_f$ of DETR-ViP trained with different $\lambda$ values on LVIS. We experiment with $\lambda = 1.0, 10.0$, and $20.0$, and observe that the best performance was achieved with $\lambda = 10.0$.

For $\tau_v$ and $\tau_t$, we conduct ablations using three values: 0.05, 0.07, and 0.1, and compare the AP on LVIS. The results in Table 8(b) show that variations in $\tau_v$ and $\tau_t$ do not lead to significant performance fluctuations. The best performance is obtained with $\tau_v = 0.1$ and $\tau_t = 0.07$. We also note that cases where $\tau_t < \tau_v$ generally yield better results, which is reasonable. Typically, we want the teacher distribution to be sharper, enabling the student to learn a less uniform distribution. However, if $\tau_t$ is too small, the teacher distribution approaches one-hot, which undermines the effect of soft labels and causes the relation distillation loss to degenerate into a supervised contrastive loss.

Table 8: Ablation Study of hyperparameters

(a) Ablation Study of $\lambda_{\text{distill}}$

| $\lambda_{\text{distill}}$ | AP | $AP_r$ | $AP_c$ | $AP_f$ |
|---|---|---|---|---|
| 1.0 | 40.5 | 30.1 | 42.2 | 41.0 |
| 10.0 | 41.1 | 35.1 | 43.3 | 40.4 |
| 20.0 | 39.8 | 34.2 | 41.2 | 39.8 |

(b) Ablation Study of $\tau_t, \tau_v$

| $\tau_v$ \ $\tau_t$ | 0.05 | 0.07 | 0.1 |
|---|---|---|---|
| 0.05 | 40.0 | 39.8 | 39.0 |
| 0.07 | 40.6 | 40.2 | 39.3 |
| 0.1 | 40.8 | 41.1 | 40.3 |

(c) Ablation Study of $\theta$

| $\theta$ | AP | $AP_r$ | $AP_c$ | $AP_f$ |
|---|---|---|---|---|
| 0.05 | 38.7 | 33.3 | 41.1 | 37.7 |
| 0.1 | 41.1 | 35.1 | 43.3 | 40.4 |
| 0.3 | 40.3 | 34.0 | 41.9 | 39.8 |
| 0.5 | 39.2 | 33.0 | 41.4 | 38.6 |

For the selective fusion strategy, we evaluate DETR-ViP under threshold values $\theta \in \{0.05, 0.1, 0.3, 0.5\}$. Intuitively, a smaller $\theta$ allows more visual prompts to participate in fusion. In this case, false negatives (FN)—categories that are present in the image but whose prompts are excluded from fusion—become less frequent. However, false positives (FP)—categories that are absent from the image but still participate in fusion—tend to increase. When $\theta$ becomes larger, the opposite trend arises: FN increases while FP decreases. At $\theta = 0.05$, DETR-ViP already achieves an AP of 38.7 on LVIS. Further analysis shows that, under $\theta = 0.05$, the number of fused prompts is reduced by nearly 75%, which effectively suppresses interference from irrelevant categories. However, when $\theta = 0.5$, AP drops significantly. We attribute this degradation to the exclusion of many in-image categories whose confidence scores are relatively low and therefore filtered out. Based on these observations, we adopt $\theta = 0.1$. This choice strikes a favorable balance: although a few irrelevant categories may still be fused, it ensures that the vast majority of categories present in the image remain included, avoiding critical FN cases while maintaining robustness.

# F    MORE ANALYSIS

## F.1    ANALYSIS OF THE FUSION MODULE

Figure 5 illustrates that the fusion strategy in Grounding DINO leads to unstable outputs when the number of prompts $N_p$ varies, while the proposed selective fusion strategy ensures robustness to prompt quantity. To demonstrate that this is not an isolated case, we additionally provide mAP-$N_p$ curves for several other categories, as shown in Figure 7. As shown in the figure, the selective fusion strategy also leads to some mAP fluctuations with varying numbers of prompts (e.g., for categories airplane and dog), but the magnitude is substantially smaller than that of the standard fusion strategy.

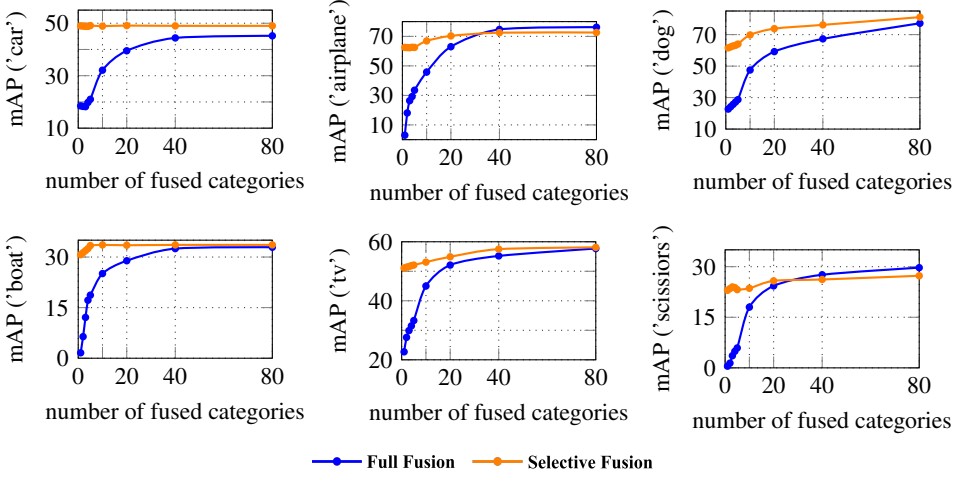

Figure 7: mAP *vs* N$_p$

Grounding DINO is also sensitive to the number of prompts. We evaluate this using the MMDetection implementations of Grounding DINO( Liu et al. (2024)) and MM Grounding DINO( Zhao et al. (2024)), which involve a critical chunked_size parameter ($L_{\text{chunked}}$). This parameter splits prompts into chunks for separate processing. As shown in Table 9, the performance of both models drops sharply when $L_{\text{chunked}}$ increases from 76 to 151, with a precise transition occurring between 80 and 95. Such instability with respect to the number of prompts not only introduces an additional hyperparameter, but also requires multiple runs under long text prompts, thereby increasing the testing time. With the selective fusion strategy, DETR-ViP integrates only the prompts corresponding to categories likely to appear in the input image, regardless of the number of user-specified prompts. This design makes it more robust and eliminates the need for chunking.

Table 9: AP w.r.t. $L$ on LVIS for Grounding DINO and MM Grounding DINO

| $L_{\text{chunked}}$ | 19 | 38 | 76 | 80 | 85 | 90 | 95 | 151 | 301 | 602 | 1203 |
|---|---|---|---|---|---|---|---|---|---|---|---|
| Grounding DINO | 29.3 | 29.0 | 28.9 | 29.7 | 29.5 | 17.9 | 6.80 | 0.26 | 0.55 | 0.35 | 0.17 |
| MM Grounding DINO | 41.2 | 43.6 | 41.3 | 40.9 | 41.0 | 25.3 | 9.34 | 0.22 | 0.57 | 0.37 | 0.18 |

## F.2    VISUAL PROMPTS IN YOLOE

As shown in Figure 1(a), the visual prompt representations in VIS-GDINO exhibit blurred and indistinct semantic boundaries. We attribute this phenomenon to the training objective used in prompt-based detection. Without contrastive image-text alignment or other auxiliary constraints to regularize the prompt feature space, visual prompts are optimized solely under the supervision of the classification loss. Moreover, under the "current image prompt, current image detect" paradigm adopted by T-Rex2 ( Jiang et al. (2024)), visual prompts are sampled only from the GT boxes of the current training image. As a result, the model observes only a limited and highly local set of object instances at each iteration, preventing the prompts from being globally optimized throughout training.

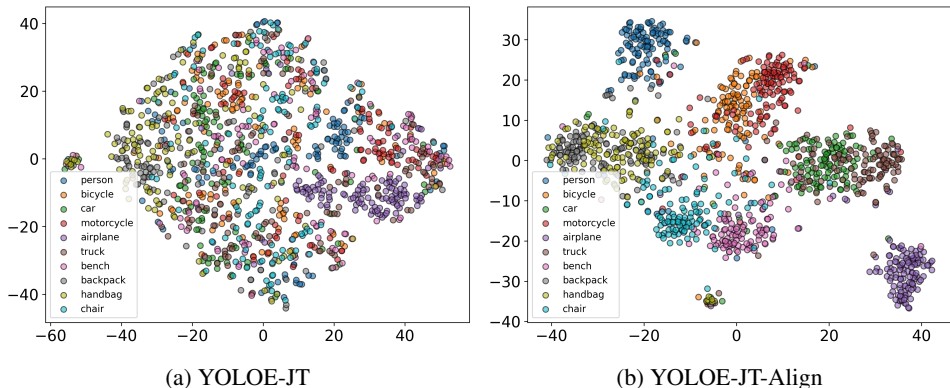

(a) YOLOE-JT           (b) YOLOE-JT-Align

Figure 8: **Visual prompt analysis for different YOLOE-JT variants.** YOLOE-JT refers to the YOLOE model obtained through joint visual-text prompt training, while YOLOE-JT-Align builds upon YOLOE-JT by incorporating an image-text prompt alignment loss.

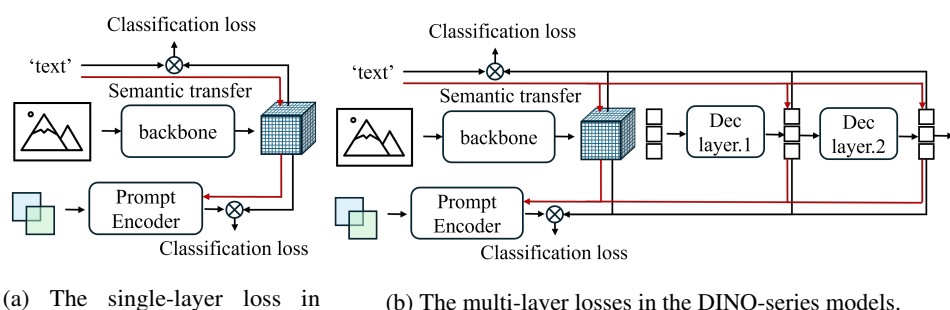

(a) The single-layer loss in YOLOE.      (b) The multi-layer losses in the DINO-series models.

Figure 9: **Classification loss and semantic transfer in YOLOE and DINO.**

To further verify this, we use the publicly available YOLOE ( Cheng et al. (2024)) code and align its training paradigm with that of T-Rex2 ( Jiang et al. (2024)), where visual-prompted detection and text-prompted detection are alternated during training. For rapid validation, we conduct experiments on YOLOE-v8s. For convenience, we denote the YOLOE model trained under this joint training scheme as **YOLOE-JT**. YOLOE-JT achieves 13.5 AP, 19.2 AP on rare categories, 13.7 on common categories, and 12.3 on frequent categories on LVIS-minival, respectively. The t-SNE visualization of its visual prompts is shown in Figure 8(a). We observe that the visual prompt distribution of YOLOE-JT appears even more scattered compared with Figure 1(a). We suspect this is because models in the DINO family adopt multi-layer losses, i.e., constraints are applied to the output of every decoder layer. As illustrated Figure 9(b), text prompts can directly optimize the multi-layer decoder outputs through multiple classification losses, and in turn, these outputs can further optimize the visual prompts. In contrast, YOLOE applies the classification loss only at the final stage of the model as shown in Figure 9(a), which results in lower efficiency in semantic propagation.

We further introduce an image-text alignment loss and denote this variant as YOLOE-JT-align. As shown in Figure 8(b), YOLOE-JT-align produces visual prompts that form more distinct category-specific clusters in the embedding space compared with YOLOE-JT. With this enhancement, YOLOE-JT-align achieves 21.5 AP, 27.8 $AP_r$, 22.2 $AP_c$, and 19.7 $AP_f$ on LVIS. These results confirm that merely adopting joint visual-text prompt training is insufficient to obtain visual prompts with coherent global semantic organization. The experiments with contrastive alignment further validate that strengthening the organization of the visual prompt space is essential for improving visual-prompted detection performance.

## F.3 VISUALIZATION

To demonstrate the effectiveness of DETR-ViP, we present extensive visualization results.

**Zero-shot Inference on COCO**    Figure 10 presents visualizations of DETR-ViP on COCO under the Visual-G protocol. For each of the 80 categories in COCO, we sampled 16 instances to extract visual prompts, and used the average of all prompts within a category as its prompt embedding.

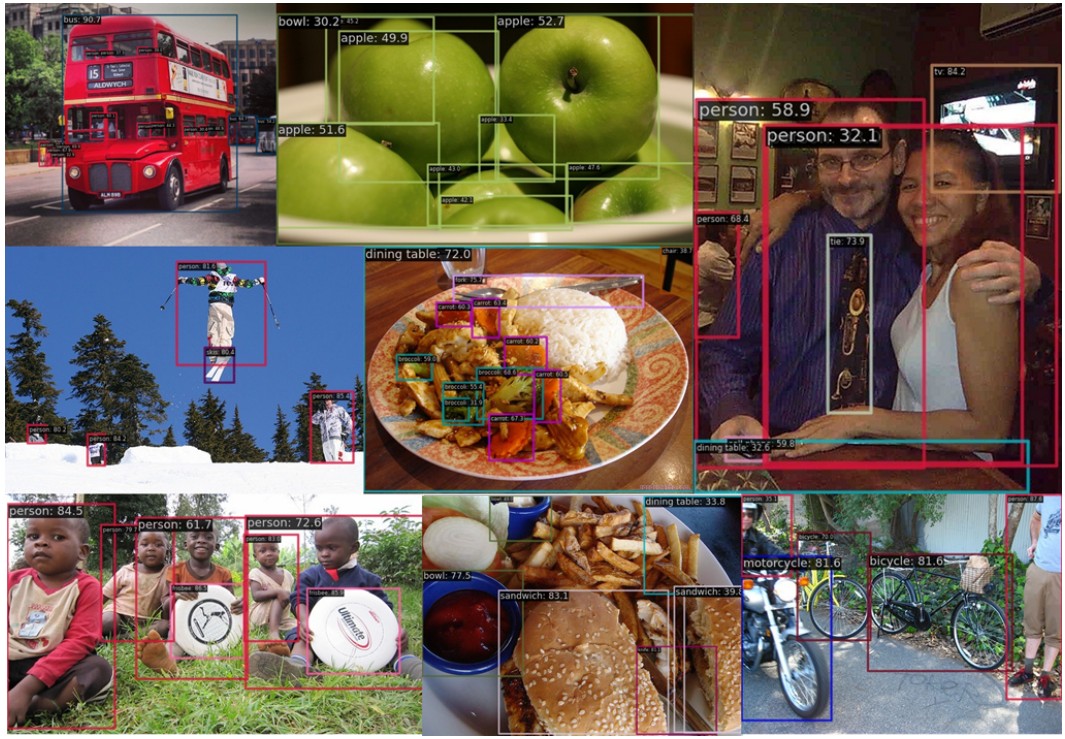

Figure 10: **Visualizations on COCO Dataset (Visual-G).**

Additionally, we provide visualizations under the Visual-I protocol in Figure 11, where bounding boxes with semi-transparent masks denote the provided visual prompts.

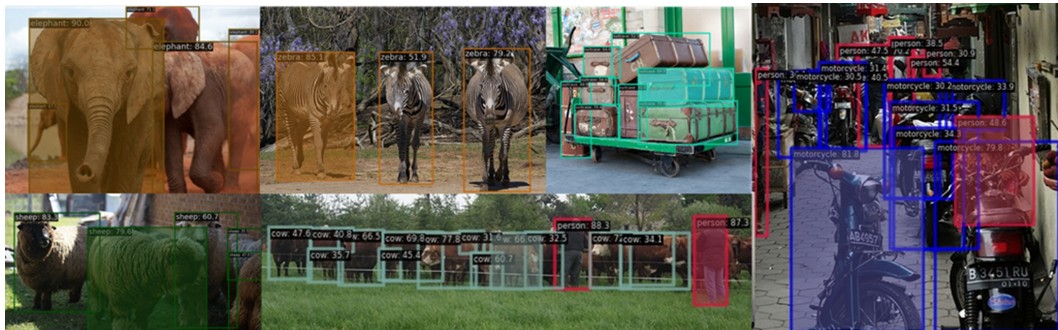

Figure 11: **Visualizations on COCO Dataset (Visual-I).**

**Zero-shot Inference on LVIS**  We showcase visualizations on the LVIS dataset under the Visual-G and Visual-I protocols in Figures 12 and 13, respectively.

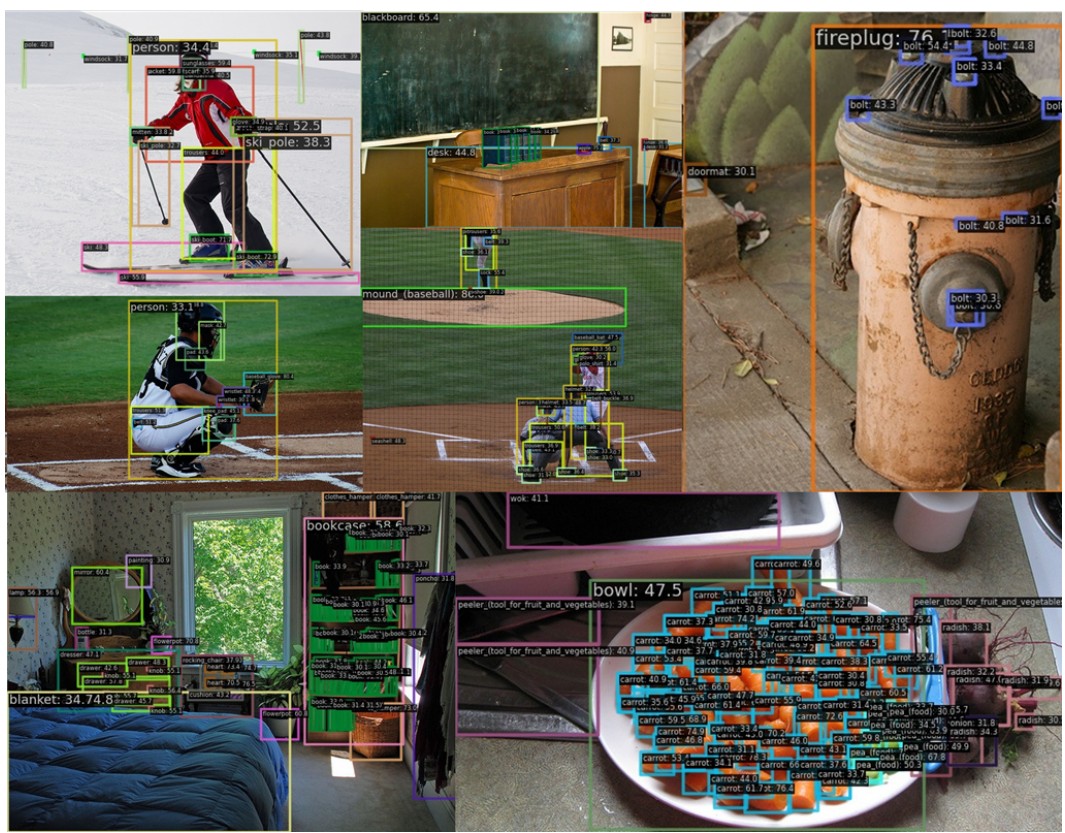

Figure 12: **Visualizations on LVIS Dataset (Visual-G).**

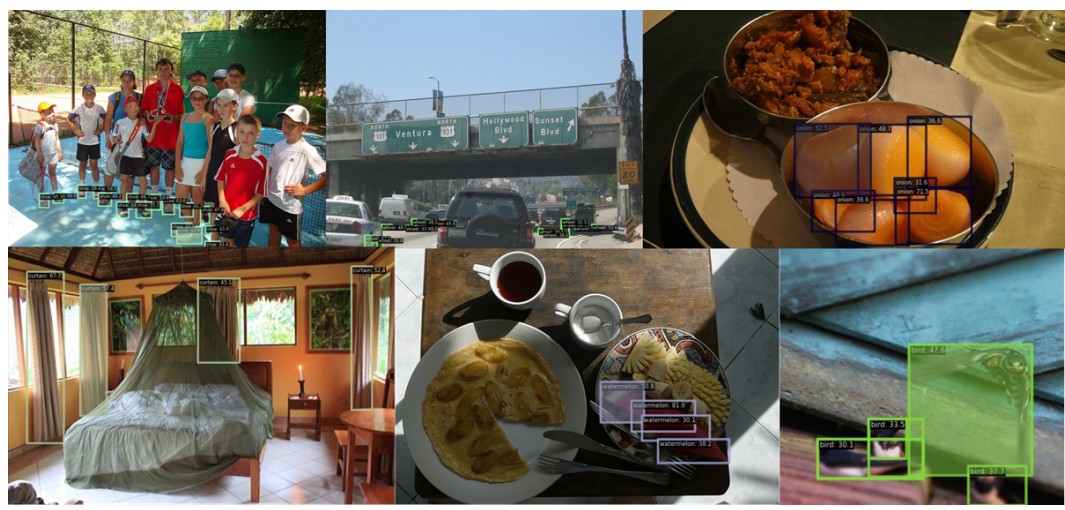

Figure 13: **Visualizations on LVIS Dataset (Visual-I).**