# OpenReview forum: "DETR-ViP: Detection Transformer with Robust Discriminative Visual Prompts"
_ICLR.cc/2026/Conference — ICLR 2026 Poster_

### Official Review · Reviewer_pcqE · 2025-10-27

**Soundness:** 4
**Presentation:** 3
**Contribution:** 3
**Rating:** 6
**Confidence:** 3

**Summary:**

This paper introduces a openset detection baseline model called DETR-ViP, The model aims to address a long-standing problem in open-vocabulary object detection: how to use visual prompts more effectively.

**Strengths:**

[1The paper is easy to follow.
[2] The ablation experiments are thorough and demonstrate the effectiveness of each module.

**Weaknesses:**

[1] The comparison experiments do not include enough existing methods for a comprehensive evaluation like dinov.
[2]  The method mainly relies on prompt ensembling and supervised contrastive learning, which are not particularly novel, and the related work section lacks corresponding discussions.

**Questions:**

[1] Can the Q-selection mechanism be effective in other transformer-based visual prompting architectures, and is the module plug-and-play?
[2] How does the method handle ambiguous targets, such as conflicts between category names and action names?
[3] Table 2 introduces a new metric, but it should clearly explain how its variation reflects the model’s performance.
[4] The zero-shot experiment only covers 10 common categories, which is insufficient to demonstrate the method’s advantage in discriminative ability.
[5] What is the difference between the distillation method in the paper and the contrastive learning used in CLIP?

---

> ### Author Response · Authors · 2025-11-20
>
> We sincerely appreciate your time and valuable comments on our work. We are pleased to see your recognition of the soundness of our approach, its contributions, the ease of following our methodology, and the comprehensiveness of our experiments. In the following responses, we address all of your concerns in detail and have revised the manuscript accordingly, with modifications highlighted in blue. We sincerely hope that these clarifications and improvements can provide us with the valuable opportunity to further raise the evaluation score of our work from your perspective.
>
> **Our Responses to Paper Weaknesses**
>
> [**W1**]: The comparison experiments do not include enough existing methods for a comprehensive evaluation like dinov.
>
> [**Response to W1**]: DINOv [1] is an excellent work that supports segmentation based on various types of prompts. However, DINOv focuses primarily on segmentation and does not evaluate general-purpose object detection under the Visual-G setting proposed in T-Rex2. **Notably, even though most of the authors of T-Rex2 are also authors of DINOv, and T-Rex2 was published later, the T-Rex2 paper still does not report DINOv’s performance on COCO and LVIS under the Visual-G setting.** For this reason, we do not include DINOv in our comparison, and instead report results only for visual-prompt detectors that publicly provide Visual-G performance—namely, T-Rex2 and YOLOE.
>
> [**W2**]: The method mainly relies on prompt ensembling and supervised contrastive learning, which are not particularly novel, and the related work section lacks corresponding discussions.
>
> [**Response to W2**]: We position the primary contribution of this work as **an investigation into why visual prompt-based detection often underperforms**, followed by empirical validation. Starting from a baseline model VIS-GDINO, we observe that visual prompts tend to lack clear semantic boundaries, which we identify as a key factor limiting their detection capability. To address this issue, we introduce global prompt integration and visual-textual relation distillation to reshape the visual prompt space, encouraging intra-class compactness and inter-class separability. As the visual prompt space becomes more semantically structured, the detection performance consistently improves, supporting our hypothesis. In addition, we extend the supervised contrastive loss by using a text-text relational matrix as soft labels, enabling better compatibility with grounding-style data.
>
> In summary, although techniques such as prompt ensembling and supervised contrastive learning themselves may not appear particularly novel, our analysis demonstrates that, motivated by the identified limitations of visual prompts, these simple mechanisms effectively enhance visual prompt-based detection. **We believe this provides useful insights for the community**.
>
> **We add a brief review of related work on contrastive learning in Appendix C to help readers who may not be familiar with this topic better understand the relevant concepts.**
>
> [1] Li, Feng, et al. "Visual in-context prompting." Proceedings of the IEEE/CVF Conference on Computer Vision and Pattern Recognition. 2024.

---

> ### Author Response · Authors · 2025-11-20
>
> **Our Responses to Questions**
>
> [**Q1**]: Can the Q-selection mechanism be effective in other transformer-based visual prompting architectures, and is the module plug-and-play?
>
> [**Response to Q1**]: I am not entirely sure what you mean by the Q-selection mechanism.
> If you are referring to the prompt-guided Q selection in Fig.2, the Q selection mechanism was originally proposed in Deformable-DETR [1] to provide better initialization for position queries. This idea has since been adopted by DN-DETR [2], DINO [3], and other subsequent works. **Our prompt-guided Q selection replaces the original scoring mechanism with the similarity between the visual prompt and image features (as in Eq.(1))**. This is conceptually similar to the language-guided query selection in Grounding DINO[4], except that the language prompt is substituted with a visual prompt.
> If you are referring to a selective fusion strategy, then our view is that as long as the model contains any kind of prompt-fusion module (regardless of the prompt modality), **this mechanism can be easily incorporated by adding a few extra classification heads—if the model already has classification heads, no additional ones are needed—together with the corresponding classification losses**.
>
> [**Q2**]: How does the method handle ambiguous targets, such as conflicts between category names and action names?
>
> [**Response to Q2**]: I'm not entirely sure what you meant by "action name." I assume you were referring to the fact that, in GoldG and similar Visual Grounding datasets, some box-aligned phrases contain verbs, such as "blades to help board cut through water" (an example come from the GoldG dataset).
> I speculate that your concern here is how to mitigate the impact of such annotations. If we strictly classify prompts based on exact phrase correspondence in global prompt integration, many samples would inevitably become false negatives.
> For such cases, **we use spaCy to extract the head noun and the lemma**, simplifying the phrase to "blade." This helps reduce (though not completely eliminate) the occurrence of false negatives.
> **We have added further details in the Appendix regarding several implementation techniques used in our global prompt integration module.**
> Furthermore, our text-visual relational distillation loss does not explicitly determine whether two phrases belong to the same class. Instead, it computes the similarity between their textual embeddings as a soft label, thereby **avoiding the problem of hard phrase matching in such cases**.
>
> [1] Zhu, Xizhou, et al. "Deformable detr: Deformable transformers for end-to-end object detection." arXiv preprint arXiv:2010.04159 (2020).
>
> [2] Li, Feng, et al. "Dn-detr: Accelerate detr training by introducing query denoising." Proceedings of the IEEE/CVF conference on computer vision and pattern recognition. 2022.
>
> [3] Zhang, Hao, et al. "Dino: Detr with improved denoising anchor boxes for end-to-end object detection." arXiv preprint arXiv:2203.03605 (2022).
>
> [4] Liu, Shilong, et al. "Grounding dino: Marrying dino with grounded pre-training for open-set object detection." European conference on computer vision. Cham: Springer Nature Switzerland, 2024.

---

> ### Author Response · Authors · 2025-11-20
>
> [**Q3**]: Table 2 introduces a new metric, but it should clearly explain how its variation reflects the model’s performance.
>
> [**Response to Q3**]: In fact, IISR is not directly or explicitly tied to model performance; instead, it is used to assess the discriminability of the visual prompt space. **We aim to demonstrate, through the aligned upward trends of both IISR and AP, our understanding of how to improve detection performance with visual prompts: a disorganized visual prompt space leads to misclassification, whereas optimizing this space enhances detection capability**.
>
> [**Q4**]: The zero-shot experiment only covers 10 common categories, which is insufficient to demonstrate the method’s advantage in discriminative ability.
>
> [**Response to Q4**]: That is a **misunderstanding**. Our experiments cover 80 classes on COCO, 804 classes on LVIS. What you refer to as the "10 common categories" applies only to the t-SNE visualization: we selected 10 categories for the plot solely to improve readability. Visualizing a much larger number of classes in a single t-SNE plot would make color distinction and interpretation difficult.
>
> [**Q5**]: What is the difference between the distillation method in the paper and the contrastive learning used in CLIP?
>
> [**Response to Q5**]: CLIP’s contrastive loss optimizes **image–text pairs** with the objective of achieving cross-modal alignment. In contrast, our distillation loss operates on **visual prompt–prompt pairs**, focusing on refining the internal structure of the visual prompt space. If visual prompts and text prompts were perfectly aligned, then the additional distillation loss would offer limited benefit. However, prior studies [1][2] have shown that **CLIP’s contrastive loss retains a modality gap**, preventing complete alignment between the visual and textual feature spaces. Under this condition, our relational distillation **bypasses the difficulty of full cross-modal alignment and directly optimizes the topology of the visual prompt space within the visual domain**.
>
> [1] Liang, Victor Weixin, et al. "Mind the gap: Understanding the modality gap in multi-modal contrastive representation learning." Advances in Neural Information Processing Systems 35 (2022): 17612-17625.
>
> [2] Schrodi, Simon, et al. "Two Effects, One Trigger: On the Modality Gap, Object Bias, and Information Imbalance in Contrastive Vision-Language Models." arXiv preprint arXiv:2404.07983 (2024).

---

> ### Author Response · Authors · 2025-11-28
>
> Dear Reviewer pcqE,
>
> We hope this message finds you well. It has now been about a week since we first submitted our response to your comments, and we would like to kindly follow up to see whether you have had a chance to look at our replies. We sincerely hope that our response has addressed the concerns you raised. If you have any further questions, we will be more than happy to provide additional clarification.
>
> Thank you again for your valuable time and constructive feedback. We truly hope that you find our work deserving of your recognition and that it may receive a favorable final assessment.
>
> Best regards,
> ﻿
>
> Authors

---

### Official Review · Reviewer_dHTA · 2025-10-31

**Soundness:** 3
**Presentation:** 3
**Contribution:** 3
**Rating:** 4
**Confidence:** 4

**Summary:**

This paper suggests the visual prompts suffer from a lack of sufficient semantic discriminability. The reseaerch question motivates the authors to propose the detection model, DETR-ViP, by enhancing the baseline model, VIS-GDINO, with more robust and discriminative visual prompts. Extensive experiments are conducted.

**Strengths:**

The paper enhances Grounding-DINO with text and visual prompts. Furthermore, three strategies are utilized to improve the visual prompts.

The experiments show that the proposed method outperforms existing detection models in the task of zero-shot generic detection.

**Weaknesses:**

In the introduction, the visualizaiton for semantic discriminability in visual prompts looks similar to VIS-GDINO, so the motivation is similar? From Figure 5 in T-Rex2, I do not find the same phenomenon in tSNE visualization. Would be nice if the authors evaluate some publicly used models to clarify the motivation.

Some parts of is a bit unclear to me. For instance, in preliminary, the $\mathcal{L}_{dn}$ in Equation 4 is not defined. Also, I couldn't find detailed description for the proposed VIS-DINO. Since VIS-DINO is also a new proposed model, the details of it can be explained?

 In the second paragraph of Introduction, the authors state "Nevertheless, visual prompts still underperform text prompts overall, which limits their practical applicability." Is there any reference or evidence for this phenomenon?

It seems the proposed method can also be used for other detection models, so would you please clarify the reason behind choosing Grounding-DINO as backbone? From Table 1 and 2, it seems Grounding-DINO can not perform better than YOLOv11 on LVIS.

**Questions:**

Please see weaknesses above.

---

> ### Author Response · Authors · 2025-11-21
>
> Thank you for your careful review and valuable suggestions, as well as your recognition of the soundness and contributions of our work. We have addressed all the concerns you raised and revised the manuscript accordingly based on your feedback. We genuinely hope that our detailed explanations  with our best efforts will give us the precious opportunity to raise the evaluation score of our work in your perspective.
>
> **Response to Paper Weaknesses**
>
> [**W2**]: Some parts of is a bit unclear to me. For instance, in preliminary, the $L_{dn}$ in Equation 4 is not defined. Also, I couldn't find detailed description for the proposed VIS-DINO. Since VIS-DINO is also a new proposed model, the details of it can be explained?
>
> [**Response to W2**]: We acknowledge that the previous writing may have caused confusion, and we apologize for that.
> The term $L_{dn}$ refers to the denoising loss introduced in DINO [1], and **we have added a clarification of its origin before Eq.(4)**. In addition, **we have expanded the descriptions of VIS-GDINO in Introduction, Method, Experiments, and Appendix D.1 to provide readers with a clearer understanding of our baseline model**.
>
> [1] Zhang, Hao, et al. "Dino: Detr with improved denoising anchor boxes for end-to-end object detection." arXiv preprint arXiv:2203.03605 (2022).

---

> ### Author Response · Authors · 2025-11-21
>
> [**W3**]: In the second paragraph of Introduction, the authors state "Nevertheless, visual prompts still underperform text prompts overall, which limits their practical applicability." Is there any reference or evidence for this phenomenon?
>
>
> [**Response to W3**]: The results of T-Rex2 on COCO and LVIS support this observation. In Appendix Table 5 (Table 4 in the initial submission), we compare the detection performance of T-Rex2 using visual prompts and text prompts, including $AP$ on COCO and $AP$, $AP_f$, and $AP_c$ on LVIS. The results indicate that, **on the majority of evaluated metrics, text prompt-based detection achieves better performance than visual prompt-based detection**.
>
>
> **Table.4 Comparison between visual and text prompts of T-Rex2 on frequent and common categories.** The results are taken from Table 1 of T-Rex2 ([1]).
>
> | Model | Prompt Type | COCO AP | LVIS AP | LVIS AP_f | LVIS AP_c |
> |-------|-------------|---------|---------|-----------|-----------|
> | T-Rex2-Swin-T | Text | 45.8 | 42.8  | 46.5 | 39.7 |
> | T-Rex2-Swin-T | Visual-G | 38.8 (↓7.0) | 37.4 (↓5.4) | 41.8 (↓4.7)  | 33.9 (↓5.8) |
> | T-Rex2-Swin-L | Text | 52.2 | 54.9  | 56.1 | 54.8  |
> | T-Rex2-Swin-L | Visual-G | 46.5 (↓5.7) | 47.6 (↓7.3) | 49.5 (↓6.6) | 46.0 (↓8.8) |
>
>
> Moreover, the T-Rex2 paper reports a similar trend: while visual prompts perform less favorably on frequent categories compared to text prompts, they demonstrate clear advantages on rare categories (see Fig. 4 in T-Rex2).
> Specially, they conducted a per-category accuracy comparison between visual and textual prompts on LVIS, as shown in Figure 4 of its original main paper. **For frequent categories, text prompts outperform visual prompts in the majority of cases (Text:Visual = 254:151)**. However, for rare categories, the reverse is more common, with visual prompts achieving higher accuracy (Text:Visual = 84:253).
>
>
> A similar observation can be found on YOLOE, as shown in the table below:
>
> **Table.4 Comparison between visual and text prompts of YOLOE on frequent and common categories.** The results are taken from Table 1 of YOLOE ([2]).
>
> | Model      | Prompt Type  | LVIS AP | LVIS AP_f | LVIS AP_c |
> |------------|--------------|---------|-----------|-----------|
> | YOLOE-v8S  | Text         |27.9     | 29.0      | 27.8      |
> | YOLOE-v8S  | Visual-G     | 26.2 (↓1.7) | 25.7(↓3.3) | 27.7(↓0.1)|
> | YOLOE-v8M | Text | 32.6 | 34.4  | 31.9 |
> | YOLOE-v8M | Visual-G | 31.0 (↓1.6) | 31.1 (↓3.3) | 31.7 (↓0.2) |
> | YOLOE-v8L | Text | 35.9 | 37.3  | 34.8 |
> | YOLOE-v8L | Visual-G | 34.2 (↓1.7) | 34.1 (↓3.2) | 34.6 (↓0.2) |
> | YOLOE-v11S | Text | 27.5 | 29.3  | 26.8 |
> | YOLOE-v11S | Visual-G | 26.3 (↓1.2) | 26.4 (↓2.9) | 27.1 (↑0.3) |
> | YOLOE-v11M | Text | 33.0 | 34.5  | 32.5 |
> | YOLOE-v11M | Visual-G | 31.4 (↓1.6) | 31.7 (↓2.8) | 31.9 (↓0.6) |
> | YOLOE-v11L | Text | 35.2 | 36.5  | 35.0 |
> | YOLOE-v11L | Visual-G | 33.7 (↓1.5) | 33.8 (↓2.7) | 34.6 (↓0.4) |
>
> [1] Jiang, Qing, et al. "T-rex2: Towards generic object detection via text-visual prompt synergy." European Conference on Computer Vision. Cham: Springer Nature Switzerland, 2024.
>
> [2] Wang, Ao, et al. "Yoloe: Real-time seeing anything." arXiv preprint arXiv:2503.07465 (2025).

---

> ### Author Response · Authors · 2025-11-21
>
> [**W4**]: It seems the proposed method can also be used for other detection models, so would you please clarify the reason behind choosing Grounding-DINO as backbone? From Table 1 and 2, it seems Grounding-DINO can not perform better than YOLOv11 on LVIS.
>
> [**Response to W4**]: We adopt Grounding DINO because our initial goal was to reproduce T-Rex2, which at that time was the only model we were aware of that supports generic detection. However, the code of T-Rex2 has not been released. Grounding DINO is developed by the same research group and follows a similar technical route, as both methods are improved variants of DINO. Although Grounding DINO does not provide official training code, it has been faithfully reimplemented in MMDetection, with a well-structured codebase and validated performance. Many components can be directly reused (e.g., ContrastiveEmbed for the language-based classification head). For these reasons, we chose Grounding DINO as the basis of our experiments.
>
> As for the model you referred to as “YOLOv11,” we assume you meant YOLOE. YOLOE is indeed a strong and well-recognized work. However, **it was accepted by ICCV 2025 on June 25, and we only became aware of it afterward**. By that time, we had already made substantial progress and completed key experiments based on Grounding DINO. Therefore, we did not migrate our work to the YOLOE framework.

---

> ### Author Response · Authors · 2025-11-25
>
> **We sincerely apologize for the late response to W1. Since we were not very familiar with the YOLOE codebase, we spent additional time studying and modifying the code. Below is our detailed response to W1.**
>
> [**W1**]:	In the introduction, the visualizaiton for semantic discriminability in visual prompts looks similar to VIS-GDINO, so the motivation is similar? From Figure 5 in T-Rex2, I do not find the same phenomenon in tSNE visualization. Would be nice if the authors evaluate some publicly used models to clarify the motivation.
>
> [**Response to W1**]: I am not entirely sure what you meant by "the visualization for semantic discriminability in visual prompts looks similar to VIS-GDINO." We suspect that this misunderstanding may stem from an unclear caption in Fig.1. To address this, **we have updated the caption to explicitly indicate that Fig.1(a) and (b) present the analysis conducted on VIS-GDINO**.
>
> Regarding the comment “From Figure 5 in T-Rex2, I do not find the same phenomenon in the t-SNE visualization,” we note that T-Rex2 has not released its full codebase (only API is available). As a result, we are unable to reproduce the visualization within their framework and cannot determine whether T-Rex2 employs undisclosed training techniques that help stabilize the visual prompt space. In contrast, when building our DETR-ViP model, we indeed observed this phenomenon and consequently developed several methods to alleviate the disorganization in the prompt embedding space.
>
> We believe this phenomenon is reasonable. VIS-GDINO is a basic baseline that does not include contrastive visual-text prompt alignment losses or other auxiliary designs. Analyzing the optimization of visual prompt in VIS-GDINO, it can be find that the supervisory signal mainly comes from the classification loss, i.e., the visual prompt embeddings sampled from the current image are compared with image features and optimized via classification loss.
> However, "under the current image prompt, current image detect" training paradigm of T-Rex2, the classification loss can only be optimized using prompts and samples from the same image, which is insufficient to inject global semantic information into visual prompts.
> Joint visual-text prompt training can implicitly transfer semantic priors from text prompts to visual prompts through:
>
> 1. **Text prompts optimizing image features**, and
> 2. **Image features guiding visual prompts via the classification loss**.
>
> Nevertheless, such knowledge transfer is indirect and relatively inefficient.
> As a result, **the visual prompts are optimized primarily at the instance level during training, without achieving globally meaningful intra-class compactness and inter-class separability**.
>
> To verify this, we modify YOLOE to adopt joint visual–text prompt training. For rapid validation, we conduct experiments on YOLOE-v8s. **Under this training paradigm, the visual prompt distribution of YOLOE similarly exhibits confusion across categories**.
> We observe that the visual prompts in this model exhibit an even more scattered distribution compared with Figure 1. We suspect that this difference arises from the training objective: the DINO family applies supervision at multiple decoder layers, where each layer is constrained by a classification loss, enabling semantic information to be repeatedly propagated from the text embeddings to the visual prompts. In contrast, YOLOE applies the classification loss only once at the final stage of the model, which reduces the efficiency of transferring semantic supervision from the text embeddings to the visual prompts.
> The model trained under this setting achieves only 13.1 AP on LVIS. We further introduce the contrastive alignment loss, which significantly improves the organization of the visual prompt space, resulting in an AP increase to 21.5.
> ***A detailed analysis, including t-SNE visualizations and experimental results, is provided in Appendix F.2 to further support our findings.***
>
> Experiments on YOLOE-v8s confirm that simply adopting joint visual–text training without introducing additional techniques is insufficient to obtain visual prompts with strong global semantic structure. Experiments with contrastive losses further demonstrate that improving the distribution of the visual prompt space is a crucial path toward enhancing visual-prompted detection performance.

---

> ### Author Response · Authors · 2025-11-27
>
> Dear Reviewer dHTA,
>
> We hope this message finds you well. It has now been about a week since we first submitted our response to your comments, and we would like to kindly follow up to see whether you have had a chance to look at our replies. We sincerely hope that our response has addressed the concerns you raised. If you have any further questions, we will be more than happy to provide additional clarification.
>
> Thank you again for your valuable time and constructive feedback. We truly hope that you find our work deserving of your recognition and that it may receive a favorable final assessment.
>
> Best regards,
>
>
> Authors

---

### Official Review · Reviewer_ZMx9 · 2025-11-01

**Soundness:** 3
**Presentation:** 2
**Contribution:** 3
**Rating:** 6
**Confidence:** 4

**Summary:**

This paper proposes DETR-ViP, a visual-prompt-based open-vocabulary detector built upon Grounding-DINO and T-Rex2. Recent studies suggest that visual prompts can outperform text prompts for rare categories, though they still underperform on common and frequent categories. Motivated by this trade-off, the method focuses on the visual-prompt regime and introduces three components to improve semantic structure and robustness: (1) Global Prompt Integration to aggregate prompts across a batch, (2) Visual–Text Relation Distillation to transfer semantic relations from language to visual prompts, and (3) Selective Fusion to suppress irrelevant prompts during feature fusion. Empirical results show competitive performance across COCO, LVIS, ODinW, and RoboFlow100 in zero-shot detection, with consistent improvements over baselines across rare, common, and frequent category groups.

**Strengths:**

- **Relevant and timely problem**

    The paper addresses an emerging direction in open-vocabulary detection, focusing on strengthening visual prompts. This is particularly relevant given recent findings that visual prompts can better support rare categories compared to textual prompts, even though they still trail on common and frequent ones. Framing the work around this trade-off makes the study well-motivated and impactful.

- **Novelty and contributions**

    The paper proposes clear and meaningful contributions toward improving the effectiveness of visual prompts for open-vocabulary detection. The introduced mechanisms are sensible, well-motivated, and grounded in practical challenges of visual-prompt learning. While the ideas are not radically complex, they are thoughtfully designed, address concrete limitations of existing approaches, and together offer an interesting and coherent advancement in visual-prompt-based detection.

- **Good empirical results**

    The evaluation covers a broad set of benchmarks (COCO, LVIS, ODinW, RoboFlow100), demonstrating consistent improvements across datasets. Importantly, gains are observed across rare, common, and frequent category subsets, strengthening the motivation for exploring visual prompts in open-vocabulary detection and supporting the practical value of the proposed design.

**Weaknesses:**

- **Method description lacks clarity in several key areas**

    Several core components are not described with sufficient precision in the main text, which makes the method harder to fully understand without forcing the reader to infer implementation behavior. In particular: the mechanics of global prompt integration remain poorly described, or the fact that selective fusion is used during training and/or inference only. Clarifying these points would greatly improve readability and transparency.
- **Fairness of comparison to T-Rex2**

    Given the architectural similarities to T-Rex2, a cleaner evaluation would require matching training conditions. However, the training data differs substantially (T-Rex2 leverages SA-1B and additional datasets, whereas DETR-ViP does not), which complicates attribution of performance gains to the proposed design alone. This point is further relevant because the T-Rex2 paper itself notes that including the SA-1B dataset “lightly weakens its generic capability.” As a result, it remains unclear to what extent differences in training data rather than architectural changes contribute to the performance gap.

- **Lack of sensitivity analysis for key hyperparameters**

    Several important hyperparameters are introduced throughout the method, including those for fusion $\lambda$, distillation strength ($λ_{distill}$), temperature parameters (τₜ, τᵥ), and focal loss weighting (α, β, γ). However, the paper does not provide sensitivity studies or robustness analysis for these choices. Since these components play a central role in shaping the behavior of the model, it would be valuable to understand how performance varies with different settings and whether the method remains stable across reasonable ranges. Including such experiments would help substantiate the robustness of the approach and clarify the extent to which reported gains depend on careful hyperparameter tuning.

**Questions:**

1. The “Global Prompt Integration” section remains very vague and the code snippet is not at the standard of top-tied venue. For example: “aggregates prompts from all samples and integrates them into a unified classifier”. What exactly are “all samples”? What does “unified classifier” refer to? My understanding is that DETR-ViP pools visual prompts from all images $\underline{\text{in the batch}}$, groups them by category, and averages them to form one prototype per class present in that batch. These batch-shared prototypes then serve as classifier weights, increasing negative diversity and stabilizing training. If this understanding is correct, I recommend describing the mechanism in this more explicit way in the paper.

2. The paper uses the $L_{dn}$ loss that appears to be the denoising loss (as in DINO). It should be explicitly stated for completeness.

3. The classification loss description does not specify whether a sigmoid is applied to similarity scores before focal loss, as done in T-Rex2. Also, are embeddings L2-normalized prior to similarity computation? Could the authors clarify the exact pipeline?

4. If visual prompts are already aligned with their corresponding text embeddings through the visual-text alignment loss as in T-Rex2, why do they not inherit from the textual semantic structure?  Can the authors comment on why is a relation-distillation loss needed in addition? Clarifying the conceptual motivation for combining both would help and strengthen the paper beyond good empirical results.

5. From my understanding, the selective fusion gating mechanism is applied during both training and inference to handle variable prompt availability and suppress irrelevant prompts. Is this correct? If so, please make it explicit in the method section.

6. Since the selective fusion strategy is motivated by interactive scenarios with variable numbers of prompts, could the authors include interactive detection and few-shot counting evaluations, similar to Tables 2–3 in T-Rex2?

7. In Table 1, T-Rex2 is trained on Object365, OpenImages, HierText, CrowdHuman and SA-1B, while DETR-ViP uses Object365 + GoldG. T-Rex2 notes SA-1B improves interactive ability but can weaken generic capability. Could the authors provide a comparison under the same training data regime? This would strengthen the paper and isolate architectural contributions.

8. I understand that the core of this paper is to improve the visual-prompt capabilities for open-vocabulary detection. But why remove text-prompt capability? Is there a practical issue in conserving this textual-prompt capability while improving the visual prompt capability? Could the authors explain why this capability was not retained?

9. Grounding DINO already includes cross-modality (image-text) feature fusion in its encoder. The ablation includes a line “+Encoder Fusion”. Is this not already present in VIS-GDINO? Please clarify what exactly is disabled/enabled.

10. In Eq. (10), the output appears to be prompt-space features (V = Wᵥ Pᵀ), but the text refers to fused image features. Could the authors adjust the description in lines 265–266 for consistency?

11. Could the authors provide sensitivity results for λ (fusion gate), α/β/γ (focal loss), τₜ/τᵥ (temperatures), and λ_distill?

12. Am I correct that the main difference from T-Rex2 in negative sampling is the global prompt integration mechanism, whereas T-Rex2 samples visual prompts within each image to avoid cross-image label inconsistencies? A clearer contrast in the " global prompt integration" section would help highlight the contribution.

---

> ### Author Response · Authors · 2025-11-20
>
> We sincerely appreciate your careful and professional review. We are pleased to see your recognition of the relevance, timeliness, and novelty of our work, as well as the substantial efforts we have made toward comprehensive experimental evaluation. We address all of your questions in detail in the following responses and will make our best effort to conduct the additional experiments you suggested, incorporating the corresponding results and revisions in the updated manuscript. We hope that our detailed explanations and forthcoming improvements can provide us with the valuable opportunity to further raise the evaluation score of our work from your perspective.
>
> **writing-related issues (W1, Q1, Q2, Q3, Q5, Q9, Q10, and Q12).**
>
> [**Response to W1**]: We have revised the description of our method in response to comments. As detailed in Response to Q1, Q12, we have improved the explanation of global prompt integration; in Response to Q5, we have refined the description of selective fusion. Beyond what was mentioned in W1, we have also added a citation for $L_\text{dn}$ (Response to Q2), clarified the similarity computation (Response to Q3), elaborated on the motivation for combining the alignment and relation distillation losses (Response to Q4), and offered a more detailed description of VIS-GDINO (Response to Q9). We sincerely appreciate your careful review, which has greatly helped us improve the quality of our manuscript.
>
> [**Response to Q1**]: Your interpretation is correct, and we have revised this part of the description in Sec 3.2 (Global Prompt Integration) accordingly.
>
> [**Response to Q2**]: Indeed, $L_\text{dn}$ corresponds to the denoising loss in DINO, and we have updated the description before Eq.(4) to explicitly clarify its origin.
>
> [**Response to Q3**]: We follow T-Rex2 in using a sigmoid function when computing the focal loss, but we do not apply L2 normalization to either the prompts or the proposals. Specially, $S=Sigmoid(OP^\top+b)$. We have updated the variable descriptions in Eq.(1), added details regarding normalization in the *Visual-Textual Prompt Relation Distillation* subsection of Sec 3.2, and provided a comprehensive clarification of all normalization-related components in Appendix D.5.
>
> [**Response to Q5**]: Your understanding is correct. We have updated the corresponding description at the end of the *Selective Fusion* subsection in Sec 3.2.
>
> [**Response to Q9**]: Yes, your understanding is correct. We have refined the description of VIS-GDINO throughout introduction, Sec 3.2, and provided additional details about VIS-GDINO in Appendix D.1.
>
> [**Response to Q10**]: The output of Eq.(10) indeed corresponds to the fused image feature produced by prompt-to-image fusion. As stated in the sentence preceding Eq.(10)—"with image-to-prompt fusion obtained by swapping $X_I$ and $P_V$"—the image-to-prompt fusion is defined as follows:
> $$
> Q=W_Q P_V, K=W_K X_I, V=W_V X_I, P_V^o = Softmax(\frac{QK^T}{\sqrt{d}})V.
> $$
>
> [**Response to Q12**]: Your understanding is correct. T-Rex2 adopts a “current image prompt, current image detect” training strategy, where visual prompts are sampled only from the ground-truth boxes of the current training image. In contrast, our global prompt integration leverages all instances of the same category within the entire batch to construct a category-level visual prompt, and concatenates the prompts of all categories in the batch to form the classifier. This design not only significantly increases the number of negative samples, but also simulates cross-image detection scenarios during training. We have added a detailed explanation of T-Rex2’s training strategy in Sec. 3.2 and compared it against our proposed global prompt integration approach.

---

> ### Author Response · Authors · 2025-11-20
>
> **The necessity of Relation Distillation Loss (Q4).**
>
> [**Q4**]: If visual prompts are already aligned with their corresponding text embeddings through the visual-text alignment loss as in T-Rex2, why do they not inherit from the textual semantic structure? Can the authors comment on why is a relation-distillation loss needed in addition? Clarifying the conceptual motivation for combining both would help and strengthen the paper beyond good empirical results.
>
>
> [**Response to Q4**]: As you noted, **if the visual–text alignment loss already aligns visual prompts with their corresponding text embeddings, the visual prompt space should, in principle, inherit the semantic structure of the text domain.** In such a scenario, the relation distillation loss would become less necessary, as its value would be small—or even approach zero—when visual prompts closely match their textual counterparts.
>
> However, existing studies show that such alignment loss alone is insufficient to achieve true cross-modal consistency. Liang et al.[1] first revealed the pervasive modality gap in multimodal models. Schrodi et al.[2] further confirmed this observation, showing that although image and text embeddings appear similar in most dimensions, they still exhibit pronounced discrepancies in several dimensions. These findings indicate that **image-text contrastive loss cannot guarantee complete alignment between visual and textual embeddings**.
>
> Unlike alignment losses that operate on visual–text pairs and require explicit cross-modal matching, **the relation-distillation loss directly optimizes the internal structure of the visual-prompt space**. It treats the relation matrix of text embeddings as a semantic prior and encourages the similarity patterns among visual prompts to follow the relation structure from the text domain. In this way, the optimization **bypasses the challenges of cross-modal alignment**, while still yielding a well-structured visual-prompt space in which prompts of the same class cluster together and those of different classes remain well separated.
>
> **Overall, the alignment loss and relation distillation loss are not conflicting; rather, they are compatible and complementary. The alignment loss provides a stable semantic anchor for optimizing visual prompts, while the relation distillation loss adjusts the topology of the visual prompt space.**
> In Section 3.2 (*Visual–Textual Prompt Relation Distillation*), we have added a description of the relationship between the two losses and the motivation for using them in combination.
>
> [1] Liang, Victor Weixin, et al. "Mind the gap: Understanding the modality gap in multi-modal contrastive representation learning." Advances in Neural Information Processing Systems 35 (2022): 17612-17625.
>
> [2] Schrodi, Simon, et al. "Two Effects, One Trigger: On the Modality Gap, Object Bias, and Information Imbalance in Contrastive Vision-Language Models." arXiv preprint arXiv:2404.07983 (2024).

---

> ### Author Response · Authors · 2025-11-20
>
> **Absence of Text Prompt Results (Q8).**
>
> [**Q8**]: I understand that the core of this paper is to improve the visual-prompt capabilities for open-vocabulary detection. But why remove text-prompt capability? Is there a practical issue in conserving this textual-prompt capability while improving the visual prompt capability? Could the authors explain why this capability was not retained?
>
> [**Response to Q8**]: We follow the cyclical training strategy of T-Rex2, which alternates between text-prompted detection and visual-prompted detection across successive iterations.
> Therefore, our model is also capable of performing text prompt-based detection.
> The results of text prompted-detection on LVIS are presented in the table below.
>
> **Table: Ablation Study on Text-Prompt Detection Accuracy Across Model Architectures**
>
> | Model                         | $AP$   | $AP_r$ | $AP_c$ | $AP_f$ |
> |-------------------------------|------|------|------|------|
> | VIS-GDINO-SwinT               | 32.0 | 26.6 | 30.3 | 34.5 |
> | +Text-Image Alignment         | 31.1 | 28.0 | 29.4 | 33.2 |
> | +Global Prompt Integration    | 31.6 | 27.1 | 29.6 | 34.1 |
> | +Text-Region Distillation     | 31.7 | 27.0 | 29.5 | 34.4 |
> | +Encoder Selective Fusion     | 27.3 | 25.3 | 24.6 | 29.4 |
> | +Decoder Selective Fusion     | 25.0 | 24.5 | 22.5 | 27.2 |
>
>
> **Although VIS-GDINO does not incorporate any modifications specifically for text prompts, it achieves an AP of 32.0 on LVIS-minival for text-prompted detection.** After introducing text-image alignment, global prompt integration, and visual-textual relation distillation, the AP shows a slight decrease but remains largely stable. However, when the fusion module is introduced, the AP for text-prompted detection drops noticeably. We hypothesize that this is due to the modality gap between visual and textual prompts, as discussed in [1][2]. When the fusion module is expected to jointly integrate both visual and textual prompts, it introduces conflicting optimization objectives, leading to a trade-off effect.
>
> Therefore, we believe that **improving the alignment between visual and textual prompts**, as well as **designing architectures that can effectively accommodate both modalities**, represents an important research direction toward building **a truly prompt-unified detector**. However, as you noted, this direction lies outside the primary focus of this work, which centers on enhancing the performance of visual prompts. For this reason, we do not further explore it in this paper. **If you believe that this discussion is necessary for the completeness of our work, we will include the corresponding exploration in the revised version.**
>
> [1] Liang, Victor Weixin, et al. "Mind the gap: Understanding the modality gap in multi-modal contrastive representation learning." Advances in Neural Information Processing Systems 35 (2022): 17612-17625.
>
> [2] Schrodi, Simon, et al. "Two Effects, One Trigger: On the Modality Gap, Object Bias, and Information Imbalance in Contrastive Vision-Language Models." arXiv preprint arXiv:2404.07983 (2024).

---

> ### Author Response · Authors · 2025-11-20
>
> **More Experiments (Q6, W2&Q7, W3&Q11).**
>
> [**Q6**]: Since the selective fusion strategy is motivated by interactive scenarios with variable numbers of prompts, could the authors include interactive detection and few-shot counting evaluations, similar to Tables 2–3 in T-Rex2?
>
> [**Response to Q6**]:
> We conduct comparisons under the Visual-I protocol. Since YOLOE does not provide results under this setting, we compare only with T-Rex2. As shown in the table, DETR-ViP consistently outperforms T-Rex2 on COCO, LVIS, ODinW, and Roboflow100 in terms of AP. We attribute this improvement to two main factors. First, **DETR-ViP learns more discriminative visual prompts**, which reinforce the model’s classification capability. The proposed global prompt integration increases the difficulty of the N-way classification task during training by exposing each category to substantially more negative samples, making the performance gain even more significant under Visual-I, where the classification space is restricted to the categories appearing in the current image. Second, **our selective fusion strategy produces prompts that better retain image-specific characteristics through more precise fusion**, thereby facilitating more accurate detection on the target image.
>
>
> **Table: Zero-shot interactive detection evaluation on COCO and LVIS.**
>
> | Model           | COCO AP | LVIS AP | LVIS $AP_f$ | LVIS $AP_c$ | LVIS $AP_r$ | ODinW $AP_{avg}$ | Roboflow100 $AP_{avg}$ |
> |-----------------|---------|---------|------------|------------|------------|---------------|-------------------|
> | T-Rex2-Swin-T   | 56.6    | 59.3    | 54.6       | 63.5       | 64.4       | 37.7          | 30.6              |
> | T-Rex2-Swin-L   | 58.5    | 62.5    | 57.9       | 66.1       | 70.1       | 39.7          | 30.2              |
> | DETR-ViP-T      | 65.4    | 66.1    | 57.5       | 73.5       | 78.4       | 46.8          | 40.1              |
> | DETR-ViP-L      | 71.1    | 71.9    | 64.2       | 78.2       | 83.6       | 51.2          | 44.3              |
>
> **We have incorporated this experiment into Sec. 4.3 of the main manuscript as one of the primary evaluations.**
>
> The experiments on object counting are still in progress.

---

> ### Author Response · Authors · 2025-11-20
>
> [**W2**]: Fairness of comparison to T-Rex2: Given the architectural similarities to T-Rex2, a cleaner evaluation would require matching training conditions. However, the training data differs substantially (T-Rex2 leverages SA-1B and additional datasets, whereas DETR-ViP does not), which complicates attribution of performance gains to the proposed design alone. This point is further relevant because the T-Rex2 paper itself notes that including the SA-1B dataset “lightly weakens its generic capability.” As a result, it remains unclear to what extent differences in training data rather than architectural changes contribute to the performance gap.
>
> [**Q7**]: In Table 1, T-Rex2 is trained on Object365, OpenImages, HierText, CrowdHuman and SA-1B, while DETR-ViP uses Object365 + GoldG. T-Rex2 notes SA-1B improves interactive ability but can weaken generic capability. Could the authors provide a comparison under the same training data regime? This would strengthen the paper and isolate architectural contributions.
>
> [**Response to W2&Q7**]:
> The reason we did not conduct experiments on the same datasets used by T-Rex2 (Objects365, OpenImages, HierText, CrowdHuman + SA-1B) is that the extremely large scale of these datasets would incur resource and time costs beyond our capacity. Therefore, we instead performed our experiments on Objects365 and GoldG.
>
> You noted that in T-Rex2, introducing the SA-1B dataset leads to a noticeable trade-off: generic detection performance decreases while interactive detection performance improves (as shown in Table 6 of T-Rex2). This observation raises a reasonable concern regarding the influence of data scale on generic detection ability—specifically, whether a model trained on Objects365 + GoldG would naturally outperform one trained on Objects365 + OpenImages + HierText + CrowdHuman, thus entangling the contributions of dataset scale and model architecture. In such a case, it would become difficult to accurately isolate and analyze the effectiveness of the mechanisms we propose.
>
> This is a reasonable and professional concern. We first analyze Table 6 in T-Rex2 to indirectly examine the impact of data scale. T-Rex2 conducts extensive ablations on the training data, including the following configurations:
>
> (1) Objects365 + GoldG (1.4M)
>
> (2) Objects365 + OpenImages + HierText + CrowdHuman (2.4M)
>
> (3) Objects365 + OpenImages + HierText + CrowdHuman + SA-1B (3.1M)
>
> Although the authors do not report the Visual-G performance of T-Rex2 trained under configuration (1), the fact that GoldG is not included in the visual-prompt training for the final model configuration may suggest that a model trained on Objects365 + GoldG did not show a clearly superior performance over configuration (2) on COCO and LVIS; otherwise, it is likely that GoldG would have been retained in the final data composition. Therefore, it is reasonable to assume that **T-Rex2 (Cfg 1) ≤ T-Rex2 (Cfg 2)**.
> On the other hand,  even when compared with the stronger configuration (2), T-Rex2 still lags behind: our DETR-ViP trained only on Objects365 + GoldG achieves 43.2 / 41.4 AP on COCO / LVIS, which is comparable to or better than the 41.1 / 38.1 obtained by T-Rex2 under configuration (2). In other words, **T-Rex2 (Cfg 2) ≤ DETR-ViP (Cfg 1)**.
>
> Although this is not definitive evidence, it provides some indication that, **on COCO and LVIS**, DETR-ViP trained on Objects365 + GoldG is unlikely to fall significantly behind T-Rex2 trained under the same data configuration.
>
> Additionally, our ablation studies, conducted under a consistent dataset (Objects365 + GoldG), progressively evaluate the contributions of each proposed module. This allows us to assess the effectiveness of our proposed mechanisms while minimizing potential confounding effects from dataset scale.
>
> We hope the above analysis can help alleviate your concerns regarding the potential impact of dataset scale. **If you still consider experiments on Objects365 + OpenImages + HierText + CrowdHuman as indispensable and critical for your evaluation of our work, we will make our best effort to request the necessary training resources to conduct the experiment.**

---

> > ### Author Response · Authors · 2025-12-02
> > **Retraining DETR-ViP on O365 + OpenImages + HierText + CrowdHuman**
> >
> > To eliminate the influence of data scale, we retrained DETR-ViP using the same data configuration as T-Rex2 (O365 + OpenImages + HierText + CrowdHuman). The results are shown in the table below:
> >
> > **Table: Comparison on LVIS across Models Trained on Different Datasets**
> > | Model      |              Dataset                | Dataset Size | $AP$ |$AP_r$|$AP_c$|$AP_f$|
> > |------------|-------------------------------------|--------------|------|------|------|------|
> > | T-Rex2-T   | O365,OpenImages,HierText,CrowdHuman |     2.4M     | 38.1 | 25.8 | 34.4 | 43.7 |
> > | DETR-ViP-T | O365,GoldG                          |     1.4M     | 41.1 | 35.1 | 43.3 | 40.4 |
> > | DETR-ViP-T | O365,OpenImages,HierText,CrowdHuman |     2.4M     | 42.4 | 28.9 | 40.7 | 44.0 |
> >
> > Compared to DETR-ViP trained on O365 + GoldG, the model trained on O365 + OpenImages + HierText + CrowdHuman achieves a higher AP of 42.4. This gain mainly comes from improvements in $AP_f$, while $AP_c$ and $AP_r$ show certain degrees of decline.
> > We attribute this behavior to the differences in dataset characteristics: datasets such as OpenImages are specifically designed for object detection and therefore provide higher-quality annotations than Visual Grounding datasets such as GoldG. Consequently, performance on frequent categories improves significantly. However, unlike Visual Grounding datasets that annotate all instances mentioned in the caption regardless of category definition, object detection datasets only annotate instances belonging to a predefined label set. This limitation leads to reduced performance on rare categories.
> > Overall, DETR-ViP trained on O365 + OpenImages + HierText + CrowdHuman still outperforms T-Rex2, demonstrating that **the mechanisms we propose effectively enhance visual-prompted detection**.

---

> ### Author Response · Authors · 2025-11-20
>
> [**W3**]: Lack of sensitivity analysis for key hyperparameters: Several important hyperparameters are introduced throughout the method, including those for fusion $\lambda$, distillation strength ($\lambda_{distill}$), temperature parameters ($\tau_t$, $\tau_v$), and focal loss weighting ($\alpha,\beta,\gamma$). However, the paper does not provide sensitivity studies or robustness analysis for these choices. Since these components play a central role in shaping the behavior of the model, it would be valuable to understand how performance varies with different settings and whether the method remains stable across reasonable ranges. Including such experiments would help substantiate the robustness of the approach and clarify the extent to which reported gains depend on careful hyperparameter tuning.
>
> [**Q11**]: Could the authors provide sensitivity results for λ (fusion gate), $\alpha,\beta,\gamma$ (focal loss), $\tau_t$, $\tau_v$ (temperatures), and $\lambda_{distill}$?
>
> [**Response to W3&Q11**]: The ablation experiments you suggested are indeed very valuable and greatly help to improve the completeness of our paper. To ensure a fair comparison with other methods, we did not modify the hyperparameters of components that are not our contributions, but instead adopted widely accepted settings. Specifically, our $\lambda_\text{cls},\lambda_\text{1},\lambda_\text{GIoU}$, and $\lambda_\text{dn}$, are kept consistent with DINO, while $\lambda_\text{Align}$ follows the setting in T-Rex2. The focal loss hyperparameters $\gamma$ and $\alpha$ are set to 2.0 and 0.25, respectively, as commonly used in previous works. The focal loss formula Eq.(5) in the initially submitted version mistakenly included an extra $\beta$, which has now been corrected.
>
> First, we conduct an ablation study on $\lambda$. The table below reports the $AP$, $AP_r$, $AP_c$, and $AP_f$ of DETR-ViP trained with different $\lambda$ values on LVIS. We experimented with $\lambda=1.0$, $10.0$, and $20.0$, and observed that the best performance was achieved with $\lambda=10.0$.
>
> **Table: Ablation Study of $\lambda_\text{distill}$**
> | $\lambda_\text{distill}$ | $AP$   | $AP_r$ | $AP_c$ | $AP_f$ |
> |------------|------|------|------|------|
> | 1.0        | 40.5 | 30.1 | 42.2 | 41.0 |
> | 10.0       | 41.1 | 35.1 | 43.3 | 40.4 |
> | 20.0       | 39.8 | 34.2 | 41.2 | 39.8 |
>
> For $\tau_v$ and $\tau_t$, we conduct ablations using three values: $0.05$, $0.07$, and $0.1$, and compare the resulting AP on LVIS. The results show that variations in $\tau_v$ and $\tau_t$ do not lead to significant performance fluctuations. The best performance is obtained with $\tau_v=0.1$ and $\tau_t=0.07$. We also note that cases where $\tau_t < \tau_v$ generally yield better results, which is reasonable. Typically, we want the teacher distribution to be sharper, **enabling the student to learn a less uniform distribution**. However, **if $\tau_t$ is too small, the teacher distribution approaches one-hot**, which undermines the effect of soft labels and causes the relation distillation loss to degenerate into a supervised contrastive loss.
>
>
> **Ablation Study of $\tau_t,\tau_v$**
> | $\tau_v$ \ $\tau_v$ | 0.05 | 0.07 | 0.1  |
> |-----------|------|------|------|
> | 0.05      | 40.0 | 39.8 | 39.0 |
> | 0.07      | 40.6 | 40.2 | 39.3 |
> | 0.1       | 40.8 | 41.1 | 40.3 |
>
> The experiments on the threshold of selective fusion have not yet been completed. In the initial version, it was denoted as $\lambda$, which could potentially be confused with the loss weights. Therefore, we now use $\theta$ to represent the threshold in selective fusion.

---

> ### Author Response · Authors · 2025-11-30
>
> **Ablation Study of $\theta$**
> | $\theta$    | $AP$   | $AP_r$| $AP_c$ | $AP_f$ |
> |------|------|------|------|------|
> | 0.05 | 38.7 | 33.3 | 41.1 | 37.7 |
> | 0.1  | 41.1 | 35.1 | 43.3 | 40.4 |
> | 0.3  | 40.3 | 34.0 | 41.9 | 39.8 |
> | 0.5  | 39.2 | 33.0 | 41.4 | 38.6 |
>
> For the selective fusion strategy, we evaluate DETR-ViP under threshold values $\theta \in \{0.05, 0.1, 0.3, 0.5\}$. Intuitively, a smaller $\theta$ allows more visual prompts to participate in fusion. In this case, false negatives (FN)—categories that are present in the image but whose prompts are excluded from fusion—become less frequent. However, false positives (FP)—categories that are absent from the image but still participate in fusion—tend to increase. When $\theta$ becomes larger, the opposite trend arises: FN increases while FP decreases. At $\theta=0.05$, DETR-ViP already achieves an AP of 38.7 on LVIS. Further analysis shows that, under $\theta=0.05$, the number of fused prompts is reduced by nearly 75\%, which effectively suppresses interference from irrelevant categories. However, when $\theta=0.5$, AP drops significantly. We attribute this degradation to the exclusion of many in-image categories whose confidence scores are relatively low and therefore filtered out. Based on these observations, we adopt $\theta=0.1$. This choice strikes a favorable balance: although a few irrelevant categories may still be fused, it ensures that the vast majority of categories present in the image remain included, avoiding critical FN cases while maintaining robustness.

---

### Author Response · Authors · 2025-11-20

We sincerely thank all reviewers for their valuable feedback, as well as their positive comments on our **meaningful research perspective** (*Reviewers ZMx9, dHTA, and pcqE*), **contributions to the community** (*Reviewers ZMx9, dHTA, and pcqE*), **good writing** (*Reviewers dHTA and pcqE*), and **extensive evaluation and comparison** (*Reviewers ZMx9, dHTA, and pcqE*).
﻿

We address all reviewers’ comments in detail below and have incorporated the corresponding revisions into the manuscript (highlighted in blue). We sincerely hope that our responses help clarify any questions or concerns raised regarding our work and contribute positively to the final evaluation. If further experiments are deemed necessary to better demonstrate the potential of DETR-ViP, we will make every effort to conduct and include them during the discussion period.

---

### Author Response · Authors · 2025-12-02
**To PCs, SACS, ACs: Summary of The Overall Rebuttal Situation. (3/3)**

**Modification of submitted documents**

We provide a summary of all modifications made to our submission, allowing PCs, SACS, ACs to easily review the changes.

I. **PDF.** Based on the reviewers’ feedback, we have carefully revised our manuscript. The major updates include:
- **Introduction**: Added a description of the baseline model VIS-GDINO.
- **Method**: Added details on similarity computation, the source of the denoising loss $L_{dn}$, the purpose and implementation of Global Prompt Integration, the distinction between relation distillation loss and alignment loss, and the stage at which the selective-fusion strategy is applied.
- **Experiments**: Added results under the Visual-I protocol.
- **Appendix D**: Added a more detailed description of the baseline model, technical details of Global Prompt Integration, and similarity computation details for all loss functions.
- **Appendix E**: Added ablation studies on hyperparameters $\lambda_{distill}$, $\tau_v,\tau_t$, and $\theta$.
- **Appendix F**: Added an analysis of visual prompts in YOLOE.

II. **Supplementary Material.** To facilitate a clearer understanding of our method for the community, we provide the relevant code in the supplementary materials, including:
- The source code of **DETR-ViP**.
- Our modifications to **YOLOE**.

**We commit to releasing our code upon the acceptance of the paper.**

**Summary**

Finally, we would like to once again express our sincere gratitude for the reviewers’ constructive feedback, which has greatly helped us improve this work. We also genuinely appreciate the additional time and effort devoted to evaluating our submission. In summary, we have addressed all raised concerns, carried out the requested experiments to the best of our ability, and carefully revised the manuscript. We hope that our summary of the rebuttal updates will help reduce the workload for the Area Chairs and assist you in making the final assessment.

Sincerely,

Authors.

---

### Author Response · Authors · 2025-12-02
**To PCs, SACS, ACs: Summary of The Overall Rebuttal Situation. (2/3)**

Next, we will summarize each reviewer's primary concerns and our detailed responses separately.

**Reviewer ZMx9**

- **Primary Concerns:**
1. (**Weakness.2**) Eliminating dataset discrepancies to isolate the effect of architectural changes.
2. (**Weakness.3**) More sensitivity analysis for key hyperparameters.
3. (**Weakness.1**) More detailed method description.
- **Our Response:**
1. We conduct all architectural ablations under the same training dataset, which effectively removes the influence of data differences and isolates the contribution of each proposed mechanism. In addition, we provide results from retraining on the O365+OpenImages+HierText+CrowdHuman configuration (due to resource and time constraints, we retrained only DETR-ViP-T). The results show that, **under identical training data, DETR-ViP still maintains a clear advantage**.
2. These experiments are indeed important for understanding our method. In our response, we provide all the ablation studies and analyses requested by the reviewers. Due to space limitations in the main text, **we have included this material in Appendix E of the revised manuscript, allowing the community to better understand our approach**.
3. Reviewer ZMx9 provided numerous suggestions for improving the quality of our writing, which are crucial for enhancing the overall quality of our manuscript. In our responses to writing-related issues (W1, Q1, Q2, Q3, Q4, Q5, Q9, Q10, and Q12), we provide detailed explanations and more comprehensive descriptions for parts of the paper that were previously unclear. **These explanations and descriptions have been incorporated into the revised manuscript, with modifications highlighted in blue**. These revisions significantly improve the readability of our paper.

**Reviewer dHTA**

- **Primary Concerns:**
1. (**Weakness.1**) Validating the phenomenon of scattered visual prompt distribution across other methods to clarify the motivation.
2. (**Weakness.3**) Providing references or evidence supporting the claim that visual prompts generally underperform textual prompts.

- **Our Response:**

1. We conducted joint visual-text prompt training on YOLOE (ICCV 2025), another detector supporting visual prompts, and similarly observed the phenomenon of scattered visual prompt distribution—indeed, it was even more pronounced than what we identified in VIS-GDINO. We attribute this issue to the training paradigm of "current image prompt, current image detect," which prevents visual prompts from being optimized at a global level. Furthermore, by applying an visual-text prompt alignment loss to YOLOE, we further verified that optimizing the visual prompt space can enhance the performance of visual-prompted detection. **We have provided a detailed description and analysis in Appendix F.2.**
2. We compare the text-prompted and visual-prompted detection results of T-Rex2 and YOLOE. The experimental results show that for frequent and common categories, text prompts generally outperform visual prompts, which supports our claim. Furthermore, a similar analysis is presented in T-Rex2 (Figure 4): among the 254 frequent categories, text prompts outperform visual prompts, whereas visual prompts outperform text prompts in only 151 categories. **A more detailed analysis is provided in Appendix B.**

**Reviewer pcqE**

- **Primary Concerns:**
1.  (**Weakness.1**) Lack of comparison with DINOv.
2.  (**Weakness.2**) Lack of novelty and related work.
- **Our Response:**
1. DINOv is a segmentation model and does not evaluate generic object detection under the Visual-G and Visual-I protocols proposed in T-Rex2. Moreover, our investigation shows that T-Rex2 is a subsequent work from the same research team as DINOv (with a large overlap in authors), and T-Rex2 itself also does not compare with DINOv. Therefore, we only compare with T-Rex2 and YOLOE. It is worth noting that **for visual-prompted detection, YOLOE, which was accepted by ICCV 2025, also compares only with T-Rex2.**
2. We believe that the core contribution of our work lies in investigating the causes of suboptimal performance in visual-prompted detection and empirically validating them using several straightforward methods. Although these methods are simple and direct, they effectively support our observations regarding the limitations of current visual-prompt training approaches, providing useful insights for the community. In addition, we include a discussion of related work on contrastive learning in Appendix C to help readers who are less familiar with this topic better understand our approach.

- **Misunderstanding:** Reviewer pcqE seems to have misunderstood that our experiments were conducted on only 10 categories. In fact, we visualized the t-SNE embeddings for the visual prompts of only 10 selected categories. This choice was made because including too many categories in the visualization would reduce color distinguishability and overall readability.

---

### Author Response · Authors · 2025-12-02
**To PCs, SACS, ACs: Summary of The Overall Rebuttal Situation. (1/3)**

Dear PCs, SACS, ACs,

We sincerely regret that the information leak incident has imposed additional workload on PCs, SACS, ACs. To facilitate your review of our rebuttals, we have compiled a summary of our responses.

We received review comments from three reviewers (ZMx9\dHTA\pcqE). **Most reviewers expressed recognition of our work**. Reviewer ZMx9 considered our work **relevant, timely, novel, and contributive**, awarding a score of 6. Reviewer pcqE found our work accessible and **easy to follow, with comprehensive and sufficient experiments**, also assigning a score of 6. Reviewer dHTA acknowledged that our method **effectively enhances visual prompts and is supported by experimental validation**. Although Reviewer dHTA assigned a score of 4, their primary concerns centered on whether the identified phenomenon is reproducible across other publicly available methods and whether evidence supports the assertion that visual prompts still underperform textual prompts. **We have addressed these questions thoroughly in our rebuttal**.

---

### Meta-Review · Area_Chair_QSiG · 2025-12-26

**Summary:**

This paper focuses on visual prompting for open-vocabulary detection via DETR-ViP, introducing global prompt integration, visual–text relation distillation, and selective fusion. The method is technically sound and empirically strong across benchmarks. It delivers clear, consistent gains, supporting acceptance.
Based on the feedback from reviewers, the decision was made to recommend it for acceptance. We congratulate the authors on their acceptance!
On the other hand, authors should revise the paper taking into account the reviewers' comments, such as the issues and concerns mentioned in Weaknesses.

**Reviewer Concerns:**

Concerns focus on clarity of method description, fairness of comparisons, limited sensitivity analysis. These issues are largely presentational or experimental scope limitations and do not undermine the core technical contributions or demonstrated effectiveness.

**Reviewer Scores:**

Reviewers report generally good-to-excellent soundness and contributions, with scores clustered around the acceptance threshold. Two reviewers lean positive (6, 6), one slightly negative (4) but open to acceptance. Overall consensus acknowledges solid empirical performance and relevance, justifying a recommendation for acceptance.

---

### Decision · Program_Chairs · 2026-01-26

Accept (Poster)